# META REINFORCEMENT LEARNING WITH AUTONOMOUS INFERENCE OF SUBTASK DEPENDENCIES

**Sungryull Sohn**[1]    **Hyunjae Woo**[1]    **Jongwook Choi**[1]    **Honglak Lee**[2,1]

[1]University of Michigan
{srsohn,hjwoo,jwook}@umich.edu

[2]Google Brain
honglak@google.com

## ABSTRACT

We propose and address a novel few-shot RL problem, where a task is characterized by a subtask graph which describes a set of subtasks and their dependencies that are unknown to the agent. The agent needs to quickly adapt to the task over few episodes during adaptation phase to maximize the return in the test phase. Instead of directly learning a meta-policy, we develop a *Meta-learner with Subtask Graph Inference* (MSGI), which infers the latent parameter of the task by interacting with the environment and maximizes the return given the latent parameter. To facilitate learning, we adopt an intrinsic reward inspired by upper confidence bound (UCB) that encourages efficient exploration. Our experiment results on two grid-world domains and StarCraft II environments show that the proposed method is able to accurately infer the latent task parameter, and to adapt more efficiently than existing meta RL and hierarchical RL methods [1].

## 1 INTRODUCTION

Recently, reinforcement learning (RL) systems have achieved super-human performance on many complex tasks (Mnih et al., 2015; Silver et al., 2016; Van Seijen et al., 2017). However, these works mostly have been focused on a single known task where the agent can be trained for a long time (*e.g.*, Silver et al. (2016)). We argue that agent should be able to solve multiple tasks with varying sources of reward. Recent work in multi-task RL has attempted to address this; however, they focused on the setting where the structure of task are *explicitly* described with natural language instructions (Oh et al., 2017; Andreas et al., 2017; Yu et al., 2017; Chaplot et al., 2018), programs (Denil et al., 2017), or graph structures (Sohn et al., 2018). However, such task descriptions may not readily be available. A more flexible solution is to have the agents infer the task by interacting with the environment. Recent work in Meta RL (Hochreiter et al., 2001; Duan et al., 2016; Wang et al., 2016; Finn et al., 2017) (especially in few-shot learning settings) has attempted to have the agents implicitly infer tasks and quickly adapt to them. However, they have focused on relatively simple tasks with a single goal (*e.g.*, multi-armed bandit, locomotion, navigation, *etc.*).

We argue that real-world tasks often have a hierarchical structure and multiple goals, which require long horizon planning or reasoning ability (Erol, 1996; Xu et al., 2017; Ghazanfari & Taylor, 2017; Sohn et al., 2018). Take, for example, the task of making a breakfast in Figure 1. A meal can be served with different dishes and drinks (*e.g.*, *boiled egg* and *coffee*), where each could be considered as a subtask. These can then be further decomposed into smaller substask until some base subtask (*e.g.*, *pickup egg*) is reached. Each subtask can provide the agent with reward; if only few subtasks provide reward, this is considered a *sparse reward* problem. When the subtask dependencies are complex and reward is sparse, learning an optimal policy can require a large number of interactions with the environment. This is the problem scope we focus on in this work: learning to quickly infer and adapt to varying hierarchical tasks with multiple goals and complex subtask dependencies.

To this end, we formulate and tackle a new few-shot RL problem called *subtask graph inference* problem, where the task is defined as a factored MDP (Boutilier et al., 1995; Jonsson & Barto, 2006) with hierarchical structure represented by *subtask graph* (Sohn et al., 2018) where the task is not known *a priori*. The task consists of multiple subtasks, where each subtask gives reward when completed (see Figure 1). The complex dependencies between subtasks (*i.e.*, preconditions) enforce agent to execute all the required subtasks *before* it can execute a certain subtask. Intuitively, the agent can efficiently solve the task by leveraging the inductive bias of underlying task structure (Section 2.2).

---

[1]The demo videos are available at https://bit.ly/msgi-videos.

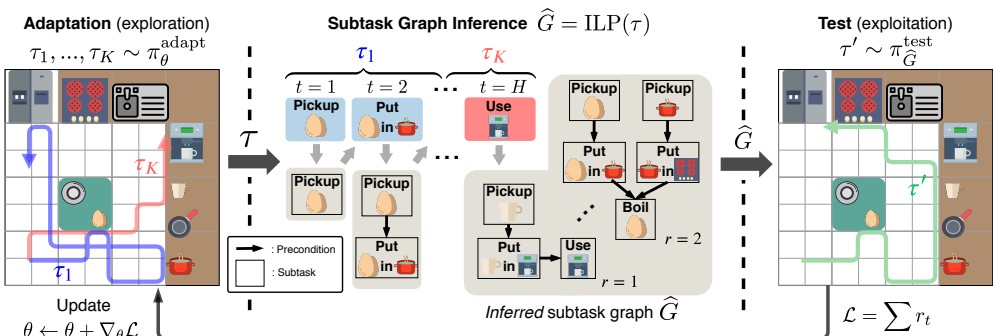

Figure 1: Overview of our method in the context of *prepare breakfast* task. This task can be broken down into subtasks (*e.g.*, *pickup mug*) that composes the underlying subtask graph $G$. (**Left**) To learn about the unknown task, the agent collects trajectories over $K$ episodes through a parameterized *adaptation* policy $\pi_\theta^{\text{adapt}}$ that learns to explore the environment. (**Center**) With each new trajectory, the agent attempts to infer the task's underlying *ground-truth* subtask graph $G$ with $\widehat{G}$. (**Right**) A separate *test* policy $\pi_{\widehat{G}}^{\text{test}}$ uses the inferred subtask graph $\widehat{G}$ to produce a trajectory that attempts to maximize the agent's reward $\sum r_t$ (*e.g.*, the green trajectory that achieves the *boil egg* subtask). The more precise $\widehat{G}$, the more reward the agent would receive, which implicitly improves the *adaptation* policy $\pi_\theta^{\text{adapt}}$ to better explore the environment and therefore better infer $\widehat{G}$ in return.

Inspired by the recent works on multi-task and few-shot RL, we propose a meta reinforcement learning approach that explicitly infers the latent structure of the task (*e.g.*, subtask graph). The agent learns its adaptation policy to collect as much information about the environment as possible in order to rapidly and accurately infer the unknown task structure. After that, the agent's test policy is a contextual policy that takes the inferred subtask graph as an input and maximizes the expected return (See Figure 1). We leverage inductive logic programming (ILP) technique to derive an efficient task inference method based on the principle of maximum likelihood. To facilitate learning, we adopt an intrinsic reward inspired by upper confidence bound (UCB) that encourages efficient exploration. We evaluate our approach on various environments ranging from simple grid-world (Sohn et al., 2018) to StarCraft II (Vinyals et al., 2017). In all cases, our method can accurately infer the latent subtask graph structure, and adapt more efficiently to unseen tasks than the baselines.

The contribution of this work can be summarized as follows:

- We propose a new meta-RL problem with more general and richer form of tasks compared to the recent meta-RL approaches.
- We propose an efficient task inference algorithm that leverages inductive logic programming, which *accurately* infers the latent subtask graph from the agent's experience data.
- We implement a deep meta-RL agent that *efficiently* infers the subtask graph for faster adaptation.
- We compare our method with other meta-RL agents on various domains, and show that our method adapts more efficiently to unseen tasks with complex subtask dependencies.

## 2 PROBLEM DEFINITION

### 2.1 BACKGROUND: FEW-SHOT REINFORCEMENT LEARNING

A *task* is defined by an MDP $\mathcal{M}_G = (\mathcal{S}, \mathcal{A}, \mathcal{P}_G, \mathcal{R}_G)$ parameterized by a task parameter $G$ with a set of states $\mathcal{S}$, a set of actions $\mathcal{A}$, transition dynamics $\mathcal{P}_G$, reward function $\mathcal{R}_G$. In the $K$-shot RL formulation (Duan et al., 2016; Finn et al., 2017), each *trial* under a fixed task $\mathcal{M}_G$ consists of an *adaptation phase* where the agent learns a task-specific behavior and a *test phase* where the adapted behavior is evaluated. For example, RNN-based meta-learners (Duan et al., 2016; Wang et al., 2016) adapt to a task $\mathcal{M}_G$ by updating its RNN states (or fast-parameters) $\phi_t$, where the initialization and update rule of $\phi_t$ is parameterized by a *slow-parameter* $\theta$: $\phi_0 = g(\theta), \phi_{t+1} = f(\phi_t; \theta)$. Gradient-based meta-learners (Finn et al., 2017; Nichol et al., 2018) instead aim to learn a good initialization of the model so that it can adapt to a new task with few gradient update steps. In the test phase, the agent's performance on the task $\mathcal{M}_G$ is measured in terms of the return:

$$\mathcal{R}_{\mathcal{M}_G}(\pi_{\phi_H}) = \mathbb{E}_{\pi_{\phi_H}, \mathcal{M}_G} \left[ \sum_{t=1}^{H'} r_t \right], \qquad (1)$$

---

**Algorithm 1** Adaptation policy optimization during meta-training

---

**Require:** $p(G)$: distribution over subtask graph
1: **while** not done **do**
2:       Sample batch of task parameters $\{G_i\}_{i=1}^M \sim p(G)$
3:       **for all** $G_i$ in the batch **do**
4:            Rollout $K$ episodes $\tau = \{\mathbf{s}_t, \mathbf{o}_t, r_t, d_t\}_{t=1}^H \sim \pi_\theta^{\text{adapt}}$ in task $\mathcal{M}_{G_i}$         $\triangleright$ adaptation phase
5:            Compute $r_t^{\text{UCB}}$ as in Eq.(7)
6:            $\widehat{G}_i = \text{ILP}(\tau)$                                    $\triangleright$ subtask graph inference
7:            Sample $\tau' \sim \pi_{\widehat{G}_i}^{\text{exe}}$ in task $\mathcal{M}_{G_i}$                      $\triangleright$ test phase
8:       Update $\theta \leftarrow \theta + \eta \nabla_\theta \sum_{i=1}^M \mathcal{R}_{\mathcal{M}_{G_i}}^{\text{PG+UCB}} \left(\pi_\theta^{\text{adapt}}\right)$ using $\mathcal{R}_{\mathcal{M}}^{\text{PG+UCB}}$ in Eq.(9)

---

where $\pi_{\phi_H}$ is the policy after $K$ episodes (or $H$ update steps) of adaptation, $H'$ is the horizon of test phase, and $r_t$ is the reward at time $t$ in the test phase. The goal is to find an optimal parameter $\theta$ that maximizes the expected return $\mathbb{E}_G[\mathcal{R}_{\mathcal{M}_G}(\pi_{\phi_H})]$ over a given distribution of tasks $p(G)$.

## 2.2 THE SUBTASK GRAPH INFERENCE PROBLEM

We formulate the *subtask graph inference* problem, an instance of few-shot RL problem where a task is parameterized by *subtask graph* (Sohn et al., 2018). The details of how a subtask graph parameterizes the MDP is described in Appendix B. Our problem extends the subtask graph *execution* problem in (Sohn et al., 2018) by removing the assumption that a subtask graph is given to the agent; thus, the agent must infer the subtask graph in order to perform the complex task. Following few-shot RL settings, the agent's goal is to quickly adapt to the given task (*i.e.*, MDP) in the adaptation phase to maximize the return in the test phase (see Figure 1). A task consists of $N$ subtasks and the subtask graph models a hierarchical dependency between subtasks.

**Subtask**: A subtask $\Phi^i$ can be defined by a tuple (*completion set* $\mathcal{S}_{\text{comp}}^i \subset \mathcal{S}$, *precondition* $G_{\mathbf{c}}^i : \mathcal{S} \mapsto \{0,1\}$, *subtask reward function* $G_{\mathbf{r}}^i : \mathcal{S} \to \mathbb{R}$). A subtask $\Phi^i$ is *complete* if the current state is contained in its completion set (*i.e.*, $\mathbf{s}_t \in \mathcal{S}_{\text{comp}}^i$), and the agent receives a reward $r_t \sim G_{\mathbf{r}}^i$ upon the completion of subtask $\Phi^i$. A subtask $\Phi^i$ is *eligible* (*i.e.*, subtask can be executed) if its precondition $G_{\mathbf{c}}^i$ is satisfied (see Figure 1 for examples). A subtask graph is a tuple of precondition and subtask reward of all the subtasks: $G = (G_{\mathbf{c}}, G_{\mathbf{r}})$. Then, the task defined by the subtask graph is a factored MDP (Boutilier et al., 1995; Schuurmans & Patrascu, 2002); *i.e.*, the transition model is factored as $p(\mathbf{s}'|\mathbf{s}, a) = \prod_i p_{G_{\mathbf{c}}^i}(s_i'|\mathbf{s}, a)$ and the reward function is factored as $R(\mathbf{s}, a) = \sum_i R_{G_{\mathbf{r}}^i}(\mathbf{s}, a)$ (see Appendix for the detail). The main benefit of factored MDP is that it allows us to model many hierarchical tasks in a principled way with a compact representation such as dynamic Bayesian network (Dean & Kanazawa, 1989; Boutilier et al., 1995). For each subtask $\Phi^i$, the agent can learn an option $\mathcal{O}^i$ (Sutton et al., 1999b) that *executes* the subtask[2].

**Environment**: The state input to the agent at time step $t$ consists of $\mathbf{s}_t = \{\mathbf{x}_t, \mathbf{e}_t, \text{step}_t, \text{epi}_t, \mathbf{obs}_t\}$.

- **Completion**: $\mathbf{x}_t \in \{0,1\}^N$ indicates whether each subtask is complete.
- **Eligibility**: $\mathbf{e}_t \in \{0,1\}^N$ indicates whether each subtask is eligible (*i.e.*, precondition is satisfied).
- **Time budget**: $\text{step}_t \in \mathbb{R}$ is the remaining time steps until episode termination.
- **Episode budget**: $\text{epi}_t \in \mathbb{R}$ is the remaining number of episodes in adaptation phase.
- **Observation**: $\mathbf{obs}_t \in \mathbb{R}^{\mathcal{H} \times \mathcal{W} \times \mathcal{C}}$ is a (visual) observation at time $t$.

At time step $t$, we denote the option taken by the agent as $\mathbf{o}_t$ and the binary variable that indicates whether episode is terminated as $d_t$.

## 3 METHOD

We propose a *Meta-learner with Subtask Graph Inference* (MSGI) which infers the latent subtask graph $G$. Figure 1 overviews our approach. Our main idea is to employ two policies: adaptation policy and test policy. During the adaptation phase, an *adaptation policy* $\pi_\theta^{\text{adapt}}$ rolls out $K$ episodes

---

[2]As in Andreas et al. (2017); Oh et al. (2017); Sohn et al. (2018), such options are pre-learned with curriculum learning; the policy is learned by maximizing the subtask reward, and the initiation set and termination condition are given as $\mathcal{I}^i = \{\mathbf{s}|G_{\mathbf{c}}^i(\mathbf{s}) = 1\}$ and $\beta^i = \mathbb{I}(x^i = 1)$

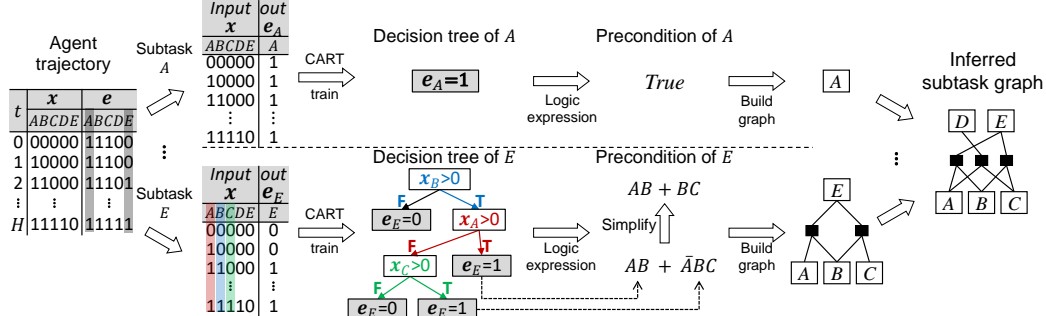

Figure 2: Our inductive logic programming module infers the precondition $G_{\mathbf{c}}$ from adaptation trajectory. For example, the decision tree of subtask $E$ (bottom row) estimates the latent precondition function $f_{G_{\mathbf{c}}^E} : \mathbf{x} \mapsto \mathbf{e}^E$ by fitting its input-output data (i.e., agent's trajectory $\{\mathbf{x}_t, \mathbf{e}_t^E\}_{t=1}^H$). The decision tree is constructed by choosing a variable (i.e., a component of $\mathbf{x}$) at each node that best splits the data. The learned decision trees of all the subtasks are represented as logic expressions, and then transformed and merged to form a subtask graph.

of *adaptation trajectories*. From the collected adaptation trajectories, the agent infers the subtask graph $\widehat{G}$ using inductive logic programming (ILP) technique. A *test policy* $\pi_{\widehat{G}}^{\text{test}}$, conditioned on the inferred subtask graph $\widehat{G}$, rolls out episodes and maximizes the return in the test phase. Note that the performance depends on the quality of the inferred subtask graph. The adaptation policy indirectly contributes to the performance by improving the quality of inference. Intuitively, if the adaptation policy completes more diverse subtasks during adaptation, the more "training data" is given to the ILP module, which results in more accurate inferred subtask graph. Algorithm 1 summarizes our meta-training procedure. For meta-testing, see Algorithm 2 in Appendix D.

### 3.1 SUBTASK GRAPH INFERENCE

Let $\tau_H = \{\mathbf{s}_1, \mathbf{o}_1, r_1, d_1, \ldots, \mathbf{s}_H\}$ be an adaptation trajectory of the adaptation policy $\pi_\theta^{\text{adapt}}$ for $K$ episodes (or $H$ steps in total) in adaptation phase. The goal is to infer the subtask graph $G$ for this task, specified by preconditions $G_{\mathbf{c}}$ and subtask rewards $G_{\mathbf{r}}$. We find the maximum-likelihood estimate (MLE) of $G = (G_{\mathbf{c}}, G_{\mathbf{r}})$ that maximizes the likelihood of the adaptation trajectory $\tau_H$: $\widehat{G}^{\text{MLE}} = \arg\max_{G_{\mathbf{c}}, G_{\mathbf{r}}} p(\tau_H | G_{\mathbf{c}}, G_{\mathbf{r}})$.

The likelihood term can be expanded as

$$p(\tau_H | G_{\mathbf{c}}, G_{\mathbf{r}}) = p(\mathbf{s}_1 | G_{\mathbf{c}}) \prod_{t=1}^H \pi_\theta\left(\mathbf{o}_t | \tau_t\right) p(\mathbf{s}_{t+1} | \mathbf{s}_t, \mathbf{o}_t, G_{\mathbf{c}}) p(r_t | \mathbf{s}_t, \mathbf{o}_t, G_{\mathbf{r}}) p(d_t | \mathbf{s}_t, \mathbf{o}_t) \quad (2)$$

$$\propto p(\mathbf{s}_1 | G_{\mathbf{c}}) \prod_{t=1}^H p(\mathbf{s}_{t+1} | \mathbf{s}_t, \mathbf{o}_t, G_{\mathbf{c}}) p(r_t | \mathbf{s}_t, \mathbf{o}_t, G_{\mathbf{r}}), \quad (3)$$

where we dropped the terms that are independent of $G$. From the definitions in Section 2.2, precondition $G_{\mathbf{c}}$ defines the mapping $\mathbf{x} \mapsto \mathbf{e}$, and the subtask reward $G_{\mathbf{r}}$ determines the reward as $r_t \sim G_{\mathbf{r}}^i$ if subtask $i$ is eligible (*i.e.*, $\mathbf{e}_t^i = 1$) and option $\mathcal{O}^i$ is executed at time $t$. Therefore, we have

$$\widehat{G}^{\text{MLE}} = (\widehat{G}_{\mathbf{c}}^{\text{MLE}}, \widehat{G}_{\mathbf{r}}^{\text{MLE}}) = \left( \arg\max_{G_{\mathbf{c}}} \prod_{t=1}^H p(\mathbf{e}_t | \mathbf{x}_t, G_{\mathbf{c}}), \ \arg\max_{G_{\mathbf{r}}} \prod_{t=1}^H p(r_t | \mathbf{e}_t, \mathbf{o}_t, G_{\mathbf{r}}) \right). \quad (4)$$

We note that no supervision from the ground-truth subtask graph $G$ is used. Below we explain how to compute the estimate of preconditions $\widehat{G}_{\mathbf{c}}^{\text{MLE}}$ and subtask rewards $\widehat{G}_{\mathbf{r}}^{\text{MLE}}$.

**Precondition inference via logic induction** Since the precondition function $f_{G_{\mathbf{c}}} : \mathbf{x} \mapsto \mathbf{e}$ (see Section 2.2 for definition) is a deterministic mapping, the probability term $p(\mathbf{e}_t | \mathbf{x}_t, G_{\mathbf{c}})$ in Eq.(4) is 1 if $\mathbf{e}_t = f_{G_{\mathbf{c}}}(\mathbf{x}_t)$ and 0 otherwise. Therefore, we can rewrite $\widehat{G}_{\mathbf{c}}^{\text{MLE}}$ in Eq.(4) as:

$$\widehat{G}_{\mathbf{c}}^{\text{MLE}} = \arg\max_{G_{\mathbf{c}}} \prod_{t=1}^H \mathbb{I}(\mathbf{e}_t = f_{G_{\mathbf{c}}}(\mathbf{x}_t)), \quad (5)$$

where $\mathbb{I}(\cdot)$ is the indicator function. Since the eligibility $\mathbf{e}$ is factored, the precondition function $f_{G_{\mathbf{c}}^i}$ for each subtask is inferred independently. We formulate the problem of finding a boolean function

that satisfies all the indicator functions in Eq.(5) (*i.e.*, $\prod_{t=1}^{H} \mathbb{I}(\mathbf{e}_t = f_{G_c}(\mathbf{x}_t)) = 1$) as an *inductive logic programming* (ILP) problem (Muggleton, 1991). Specifically, $\{\mathbf{x}_t\}_{t=1}^{H}$ forms binary vector inputs to programs, and $\{e_t^i\}_{t=1}^{H}$ forms Boolean-valued outputs of the $i$-th program that denotes the eligibility of the $i$-th subtask. We use the *classification and regression tree* (CART) to infer the precondition function $f_{G_c}$ for each subtask based on Gini impurity (Breiman, 1984). Intuitively, the constructed decision tree is the simplest boolean function approximation for the given input-output pairs $\{\mathbf{x}_t, \mathbf{e}_t\}$. Then, we convert it to a logic expression (*i.e.*, precondition) in sum-of-product (SOP) form to build the subtask graph. Figure 2 summarizes the overall logic induction process.

**Subtask reward inference**   To infer the subtask reward function $\widehat{G}_{\mathbf{r}}^{\text{MLE}}$ in Eq.(4), we model each component of subtask reward as a Gaussian distribution $G_{\mathbf{r}}^i \sim \mathcal{N}(\widehat{\mu}^i, \widehat{\sigma}^i)$. Then, $\widehat{\mu}_{\text{MLE}}^i$ becomes the empirical mean of the rewards received after taking the eligible option $\mathcal{O}^i$ in the trajectory $\tau_H$:

$$\widehat{G}_{\mathbf{r}}^{\text{MLE},i} = \widehat{\mu}_{\text{MLE}}^i = \mathbb{E}\left[r_t | \mathbf{o}_t = \mathcal{O}^i, \mathbf{e}_t^i = 1\right] = \frac{\sum_{t=1}^{H} r_t \mathbb{I}(\mathbf{o}_t = \mathcal{O}^i, \mathbf{e}_t^i = 1)}{\sum_{t=1}^{H} \mathbb{I}(\mathbf{o}_t = \mathcal{O}^i, \mathbf{e}_t^i = 1)}. \qquad (6)$$

### 3.2   Test phase: Subtask Graph Execution Policy

Once a subtask graph $\widehat{G}$ has been inferred, we can derive a *subtask graph execution (SGE) policy* $\pi_{\widehat{G}}^{\text{exe}}(\mathbf{o}|\mathbf{x})$ that aims to maximize the cumulative reward in the test phase. Note that this is precisely the problem setting used in Sohn et al. (2018). Therefore, we employ a graph reward propagation (GRProp) policy (Sohn et al., 2018) as our SGE policy. Intuitively, the GRProp policy approximates a subtask graph to a differentiable form such that we can compute the gradient of modified return with respect to the completion vector to measure how much each subtask is likely to increase the modified return. Due to space limitation, we give a detail of the GRProp policy in Appendix I.

### 3.3   Learning: Optimization of the Adaptation Policy

We now describe how to learn the adaptation policy $\pi_\theta^{\text{adapt}}$, or its parameters $\theta$. We can directly optimize the objective $\mathcal{R}_{\mathcal{M}_G}(\pi)$ using policy gradient methods (Williams, 1992; Sutton et al., 1999a), such as actor-critic method with generalized advantage estimation (GAE) (Schulman et al., 2016). However, we find it challenging to train our model for two reasons: 1) delayed and sparse reward (*i.e.*, the return in the test phase is treated as if it were given as a one-time reward at the last step of adaptation phase), and 2) large task variance due to highly expressive power of subtask graph. To facilitate learning, we propose to give an intrinsic reward $r_t^{\text{UCB}}$ to agent in addition to the extrinsic environment reward, where $r_t^{\text{UCB}}$ is the upper confidence bound (UCB) (Auer et al., 2002)-inspired exploration bonus term as follows:

$$r_t^{\text{UCB}} = w_{\text{UCB}} \cdot \mathbb{I}(\mathbf{x}_t \text{ is novel}), \quad w_{\text{UCB}} = \sum_{i=1}^{N} \frac{\log(n^i(0) + n^i(1))}{n^i(e_t^i)}, \qquad (7)$$

where $N$ is the number of subtasks, $e_t^i$ is the eligibility of subtask $i$ at time $t$, and $n^i(e)$ is the visitation count of $e^i$ (*i.e.*, the eligibility of subtask $i$) during the adaptation phase until time $t$. The weight $w_{\text{UCB}}$ is designed to encourage the agent to make eligible and execute those subtasks that have infrequently been eligible, since such rare data points in general largely improve the inference by balancing the dataset that CART (*i.e.*, our logic induction module) learns from. The conditioning term $\mathbb{I}(\mathbf{x}_t \text{ is novel})$ encourages the adaptation policy to visit novel states with a previously unseen completion vector $\mathbf{x}_t$ (*i.e.*, different combination of completed subtasks), since the data points with same $\mathbf{x}_t$ input will be ignored in the ILP module as a duplication. We implement $\mathbb{I}(\mathbf{x}_t \text{ is novel})$ using a hash table for computational efficiency. Then, the intrinsic objective is given as follows:

$$\mathcal{R}_{\mathcal{M}_G}^{\text{UCB}}\left(\pi_\theta^{\text{adapt}}\right) = \mathbb{E}_{\pi_\theta^{\text{adapt}}, \mathcal{M}_G}\left[\sum_{t=1}^{H} r_t^{\text{UCB}}\right], \qquad (8)$$

where $H$ is the horizon of adaptation phase. Finally, we train the adaptation policy $\pi_\theta^{\text{adapt}}$ using an actor-critic method with GAE (Schulman et al., 2016) to maximize the following objective:

$$\mathcal{R}_{\mathcal{M}_G}^{\text{PG+UCB}}\left(\pi_\theta^{\text{adapt}}\right) = \mathcal{R}_{\mathcal{M}_G}\left(\pi_{\widehat{G}}^{\text{GRProp}}\right) + \beta_{\text{UCB}} \mathcal{R}_{\mathcal{M}_G}^{\text{UCB}}\left(\pi_\theta^{\text{adapt}}\right), \qquad (9)$$

where $\mathcal{R}_{\mathcal{M}_G}(\cdot)$ is the meta-learning objective in Eq.(1), $\beta_{\text{UCB}}$ is the mixing hyper-parameter, and $\widehat{G}$ is the inferred subtask graph that depends on the adaptation policy $\pi_\theta^{\text{adapt}}$. The complete procedure for training our MSGI agent with UCB reward is summarized in Algorithm 1.

## 4 RELATED WORK

**Meta Reinforcement Learning.** There are roughly two broad categories of meta-RL approaches: gradient-based meta-learners (Finn et al., 2017; Nichol et al., 2018; Gupta et al., 2018; Finn et al., 2018; Kim et al., 2018) and RNN-based meta-learners (Duan et al., 2016; Wang et al., 2016). Gradient-based meta RL algorithms, such as MAML (Finn et al., 2017) and Reptile (Nichol et al., 2018), learn the agent's policy by taking policy gradient steps during an adaptation phase, where the meta-learner aims to learn a good initialization that enables rapid adaptation to an unseen task. RNN-based meta-RL methods (Duan et al., 2016; Wang et al., 2016) updates the hidden states of a RNN as a process of adaptation, where both of hidden state initialization and update rule are meta-learned. Other variants of adaptation models instead of RNNs such as temporal convolutions (SNAIL) (Mishra et al., 2018) also have been explored. Our approach is closer to the second category, but different from existing works as we directly and explicitly infer the task parameter.

**Logic induction.** Inductive logic programming systems (Muggleton, 1991) learn a set of rules from examples. (Xu et al., 2017) These works differ from ours as they are open-loop LPI; the input data to LPI module is generated by other policy that does not care about ILP process. However, our agent learns a policy to collect data more efficiently (i.e., closed-loop ILP). There also have been efforts to combine neural networks and logic rules to deal with noisy and erroneous data and seek data efficiency, such as (Hu et al., 2016; Evans & Grefenstette, 2017; Dong et al., 2019).

**Autonomous Construction of Task Structure.** Task planning approaches represented the task structure using Hierarchical Task Networks (HTNs) (Tate, 1977). HTN identifies subtasks for a given task and represent symbolic representations of their preconditions and effects, to reduce the search space of planning (Hayes & Scassellati, 2016). They aim to execute a single goal task, often with assumptions of simpler subtask dependency structures (*e.g.*, without *NOT* dependency (Ghazanfari & Taylor, 2017; Liu et al., 2016)) such that the task structure can be constructed from the successful trajectories. In contrast, we tackle a more general and challenging setting, where each subtask gives a reward (*i.e.*, multi-goal setting) and the goal is to maximize the cumulative sum of reward within an episode. More recently, these task planning approaches were successfully applied to the few-shot visual imitation learning tasks by constructing recursive programs (Xu et al., 2017) or graph (Huang et al., 2018). Contrary to them, we employ an *active* policy that seeks for experience useful in discovering the task structure in unknown and stochastic environments.

## 5 EXPERIMENTS

In the experiment, we investigate the following research questions: (1) Does MSGI correctly infer task parameters $G$? (2) Does adaptation policy $\pi_\theta^{\text{adapt}}$ improve the efficiency of few-shot RL? (3) Does the use of UCB bonus facilitate training? (See Appendix H.1) (4) How well does MSGI perform compared with other meta-RL algorithms? (5) Can MSGI generalize to longer adaptation horizon, and unseen and more complex tasks?

We evaluate our approach in comparison with the following baselines:

- Random is a policy that executes a random eligible subtask that has not been completed.
- $\text{RL}^2$ is the meta-RL agent in Duan et al. (2016), trained to maximize the return over $K$ episodes.
- HRL is the hierarchical RL agent in Sohn et al. (2018) trained with the same actor-critic method as our approach during adaptation phase. The network parameter is reset when the task changes.
- GRProp+Oracle is the GRProp policy (Sohn et al., 2018) provided with the ground-truth subtask graph as input. This is roughly an upper bound of the performance of MSGI-based approaches.
- MSGI-Rand (Ours) uses a random policy as an adaptation policy, with the task inference module.
- MSGI-Meta (Ours) uses a meta-learned policy (*i.e.*, $\pi_\theta^{\text{adapt}}$) as an adaptation policy, with the task inference module.

For $\text{RL}^2$ and HRL, we use the same network architecture as our MSGI adaptation policy. More details of training and network architecture can be found in Appendix J. The domains on which we evaluate these approaches include two simple grid-world environments (**Mining** and **Playground**) (Sohn et al., 2018) and a more challenging domain **SC2LE** (Vinyals et al., 2017) (StarCraft II).

### 5.1 EXPERIMENTS ON MINING AND PLAYGROUND DOMAINS

**Mining** (Sohn et al., 2018) is inspired by Minecraft (see Figure 3) where the agent receives reward by picking up raw materials in the world or crafting items with raw materials. **Playground** (Sohn et al.,

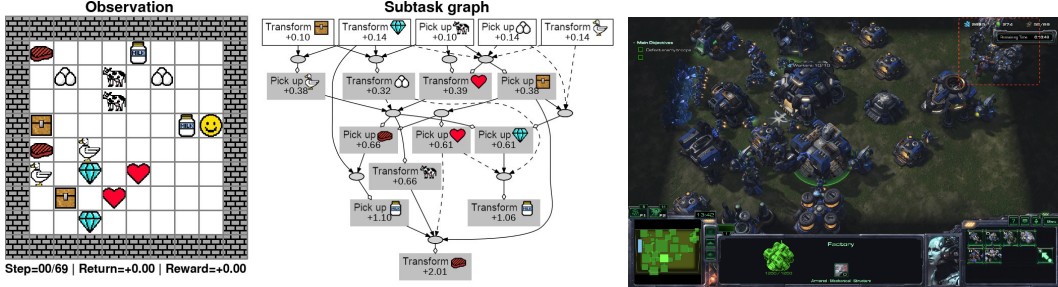

Figure 3: **Left**: A visual illustration of **Playground** domain and an example of underlying subtask graph. The goal is to execute subtasks in the optimal order to maximize the reward within time budget. The subtask graph describes subtasks with the corresponding rewards (e.g., transforming a chest gives 0.1 reward) and dependencies between subtasks through AND and OR nodes. For instance, the agent must first *transform chest* AND *transform diamond* before executing *pick up duck*. **Right**: A *warfare* scenario in **SC2LE** domain (Vinyals et al., 2017). The agent must prepare for the upcoming warfare by training appropriate units, through an appropriate order of subtasks (see Appendix for more details).

2018) is a more flexible and challenging domain, where the environment is stochastic and subtask graphs are *randomly* generated (*i.e.*, precondition is an arbitrary logic expression). We follow the setting in Sohn et al. (2018) for choosing train/evaluation sets. We measure the performance in terms of normalized reward $\widehat{R} = (R - R_{\min})/(R_{\max} - R_{\min})$ averaged over 4 random seeds, where $R_{\min}$ and $R_{\max}$ correspond to the average reward of the Random and the GRProp+Oracle agent, respectively.

### 5.1.1 TRAINING PERFORMANCE

Figure 4 shows the learning curves of MSGI-Meta and RL$^2$, trained on the **D1**-Train set of **Playground** domain. We set the adaptation budget in each trial to $K = 10$ episodes. For MSGI-Rand and HRL (which are not meta-learners), we show the average performance after 10 episodes of adaptation. As training goes on, the performance of MSGI-Meta significantly improves over MSGI-Rand with a large margin. It demonstrates that our meta adaptation policy learns to explore the environment more efficiently, inferring subtask graphs more accurately. We also observe that the performance of RL$^2$ agent improves over time, eventually outperforming the HRL agent. This indicates that RL$^2$ learns 1) a good initial policy parameter that captures the common knowledge generally applied to all the tasks and 2) an efficient adaptation scheme such that it can adapt to the given task more quickly than standard policy gradient update in HRL.

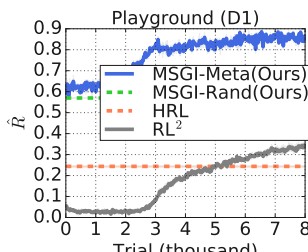

Figure 4: Learning curves on the **Playground** domain. We measure the normalized reward (y-axis) in a test phase, after a certain number of training trials (x-axis).

### 5.1.2 ADAPTATION AND GENERALIZATION PERFORMANCE

**Adaptation efficiency.** In Figure 5, we measure the test performance (in terms of the normalized reward $\widehat{R}$) by varying episode budget $K$ (*i.e.*, how many episodes are used in adaptation phase), after 8000 trials of meta-training (Figure 4). Intuitively, it shows how quickly the agent can adapt to the given task. Our full algorithm MSGI-Meta consistently outperforms MSGI-Rand across all the tasks, showing that our meta adaptation policy can efficiently explore informative states that are likely to result in more accurate subtask graph inference. Also, both of our MSGI-based models perform better than HRL and RL$^2$ baselines in all the tasks, showing that explicitly inferring underlying task structure and executing the predicted subtask graph is more effective than learning slow-parameters and fast-parameters (*e.g.*, RNN states) on those tasks involving complex subtask dependencies.

**Generalization performance.** We test whether the agents can generalize over unseen task and longer adaptation horizon, as shown in Figure 5. For **Playground**, we follow the setup of (Sohn et al., 2018): we train the agent on **D1**-Train with the adaptation budget of 10 episodes, and test on unseen graph distributions **D1**-Eval and larger graphs **D2-D4** (See Appendix C for more details about the tasks in Playground and Mining). We report the agent's performance as the normalized reward with up to 20 episodes of adaptation budget. For **Mining**, the agent is trained on randomly generated graphs with 25 episodes budget and tested on 440 hand-designed graphs used in (Sohn et al., 2018), with up to

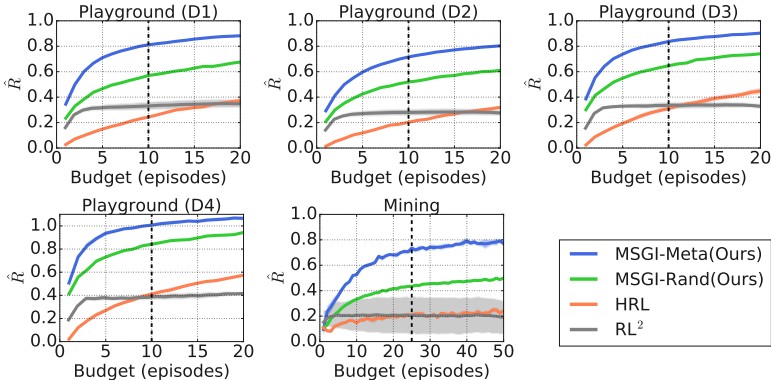

Figure 5: Generalization performance on unseen tasks (**D1**-Eval, **D2, D3, D4**, and **Mining**-Eval) with varying adaptation horizon. We trained agent with the fixed adaptation budget ($K = 10$ for **Playground** and $K = 25$ for **Mining**) denoted by the vertical dashed line, and tested with varying unseen adaptation budgets. We report the average normalized return during test phase, where GRProp+Oracle is the upper bound (*i.e.*, $\widehat{R} = 1$) and Random is the lower bound (*i.e.*, $\widehat{R} = 0$). The shaded area in the plot indicates the range between $\widehat{R} + \sigma$ and $\widehat{R} - \sigma$ where $\sigma$ is the standard error of normalized return.

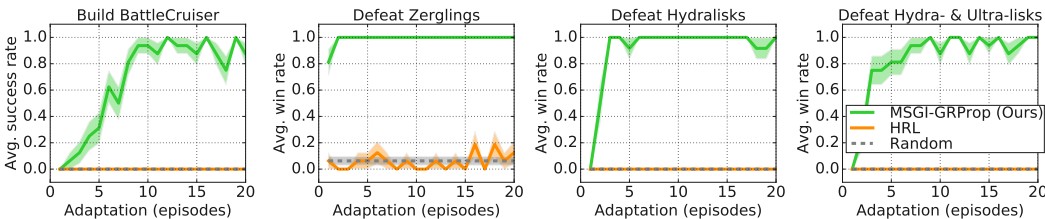

Figure 6: Adaptation performance with different adaptation horizon on **SC2LE** domain.

50 episodes of adaptation budget. Both of our MSGI-based models generalize well to unseen tasks and over different adaptation horizon lengths, continuingly improving the agent's performance. It demonstrates that the efficient exploration scheme that our meta adaptation policy can generalize to unseen tasks and longer adaptation horizon, and that our task execution policy, GRProp, generalizes well to unseen tasks as already shown in (Sohn et al., 2018). However, $\text{RL}^2$ fails to generalize to unseen task and longer adaptation horizon: on **D2-D4** with adaptation horizons longer than the length the meta-learner was trained for, the performance of the $\text{RL}^2$ agent is almost stationary or even decreases for very long-horizon case (**D2, D3**, and **Mining**), eventually being surpassed by the HRL agent. This indicates (1) the adaptation scheme that $\text{RL}^2$ learned does not generalize well to longer adaptation horizons, and (2) a common knowledge learned from the training tasks does not generalize well to unseen test tasks.

## 5.2 EXPERIMENTS ON STARCRAFT II DOMAIN

**SC2LE** (Vinyals et al., 2017) is a challenging RL domain built upon the real-time strategy game StarCraft II. We focus on two particular types of scenarios: *Defeat Enemy* and *Build Unit*. Each type of the scenarios models the different aspect of challenges in the full game. The goal of *Defeat Enemy* is to eliminate various enemy armies invading within 2,400 steps. We consider three different combinations of units with varying difficulty: *Defeat Zerglings, Defeat Hydralisks, Defeat Hydralisks & Ultralisks* (see Figure 9 and demo videos at https://bit.ly/msgi-videos). The goal of *Build Unit* scenario is to build a specific unit within 2,400 steps. To showcase the advantage of MSGI inferring the underlying subtask graph, we set the target unit as *Battlecruiser*, which is at the highest rank in the technology tree of *Terran* race. In both scenarios, the agent needs to train the workers, collect resources, and construct buildings and produce units in correct sequential order to win the game. Each building or unit has a precondition as per the technology tree of the player's race (see Figure 11 and Appendix E for more details).

**Agents.** Note that the precondition of each subtask is determined by the domain and remains fixed across the tasks. If we train the meta agents (MSGI-Meta and RL$^2$), the agents memorize the subtask dependencies (*i.e.*, over-fitting) and does not learn any useful policy for efficient adaptation. Thus, we only evaluate Random and HRL as our baseline agents. Instead of MSGI-Meta, we used MSGI-GRProp. MSGI-GRProp uses the GRProp policy as an adaptation policy since GRProp is a good approximation algorithm that works well without meta-training as shown in (Sohn et al., 2018). Since the environment does not provide any subtask-specific reward, we set the subtask reward using the UCB bonus term in Eq. (7) to encourage efficient exploration (See Appendix for detail).

**Subtask graph inference.** We quantitatively evaluate the inferred subtask graph in terms of the precision and recall of the inferred precondition function $f_{\widehat{e}} : x \mapsto \widehat{e}$. Specifically, we compare the inference output $\widehat{e}$ with the GT label $e$ generated by the GT precondition function $f_e : x \mapsto e$ for all possible binary assignments of input (i.e., completion vector $x$). For all the tasks, our MSGI-GRProp agent almost perfectly infers the preconditions with more than 94% precision and 96% recall of all possible binary assignments, when averaged over all 163 preconditions in the game, with only 20 episodes of adaptation budget. We provide the detailed quantitative and qualitative results on the inferred subtask graph in supplemental material.

**Adaptation efficiency.** Figure 6 shows the adaptation efficiency of MSGI-GRProp, HRL agents, and Random policy on the four scenarios. We report the average victory or success rate over 8 episodes. MSGI-GRProp consistently outperforms HRL agents with a high victory rate, by (1) quickly figuring out the useful units and their prerequisite buildings and (2) focusing on executing these subtasks in a correct order. For example, our MSGI-GRProp learns from the inferred subtask graph that some buildings such as *sensor tower* or *engineering bay* are unnecessary for training units and avoids constructing them (see Appendix F for the inferred subtask graph).

## 6 CONCLUSION

We introduced and addressed a few-shot RL problem with a complex subtask dependencies. We proposed to learn the adaptation policy that efficiently collects experiences in the environment, infer the underlying hierarchical task structure, and maximize the expected reward using the execution policy given the inferred subtask graph. The empirical results confirm that our agent can efficiently explore the environment during the adaptation phase that leads to better task inference and leverage the inferred task structure during the test phase. In this work, we assumed that the option is pre-learned and the environment provides the status of each subtask. In the future work, our approach may be extended to more challenging settings where the relevant subtask structure is fully learned from pure observations, and options to execute these subtasks are also automatically discovered.

ACKNOWLEDGMENTS

We would like to thank Wilka Carvalho for valuable feedback on the manuscript. This work was partly supported by Institute for Information & communications Technology Promotion (IITP) grant funded by the Korea government (MSIT) (No. 2016-0-00563, *Research on Adaptive Machine Learning Technology Development for Intelligent Autonomous Digital Companion*) and Korea Foundation for Advanced Studies.

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

# Appendix: Meta Reinforcement Learning with Autonomous Inference of Subtask Dependencies

## A  SUBTASK GRAPH AND FACTORED MDP

### A.1  BACKGROUND: FACTORED MARKOV DECISION PROCESSES

A factored MDP (FMDP) (Boutilier et al., 1995; Jonsson & Barto, 2006) is an MDP $\mathcal{M} = (\mathcal{S}, \mathcal{A}, \mathcal{P}, \mathcal{R})$, where the state space $\mathcal{S}$ is defined by a set of discrete state variables $\mathbf{s} = \{s^1, \ldots, s^d\}$. Each state variable $s^i \in \mathbf{s}$ takes on a value in its domain $\mathcal{D}(s^i)$. The state set $\mathcal{S}$ is a (subset of) Cartesian product of the domain of all state variables $\times_{s^i \in \mathbf{s}} \mathcal{D}(s^i)$. In FMDP, the state variables $s^i$ are conditionally independent, such that the transition probability can be factored as follows:

$$p(\mathbf{s}_{t+1}|\mathbf{s}_t, a_t) = p(s_{t+1}^1|\mathbf{s}_t, a_t)p(s_{t+1}^2|\mathbf{s}_t, a_t) \ldots p(s_{t+1}^d|\mathbf{s}_t, a_t). \tag{10}$$

Then, the model of FMDP can be compactly represented by the subtask graph (Sohn et al., 2018) or dynamic Bayesian network (DBN) (Dean & Kanazawa, 1989; Boutilier et al., 1995). They represent the transition of each state variable $p(s_{t+1}^i|\mathbf{s}_t, a_t)$ in either a Boolean expression (*i.e.*, subtask graph) or a binary decision tree (*i.e.*, DBN). For more intuitive explanation, see the **subtask graph** paragraph in Section 2.2 and Figure 1.

Jonsson & Barto (2006); Sohn et al. (2018) suggested that the factored MDP can be extended to the option framework. Specifically, the option is defined based on the change in state variable (*e.g.*, completion of subtask in Sohn et al. (2018)), and the option transition model and option reward function are assumed to be factored. Similar to Eq. 10, the transition probability can be factored as follows:

$$p(\mathbf{s}'|\mathbf{s}, o) = \prod_i p(s_i'|\mathbf{s}, o), \qquad R(\mathbf{s}, o) = \sum_i R^i(\mathbf{s}, o). \tag{11}$$

In (Sohn et al., 2018), the option $\mathcal{O}^i$ completes the subtask $\Phi^i$ by definition; thus, $p(s_i'|\mathbf{s}, o) = 0$ and $R^i(\mathbf{s}, o) = 0$ *if* $o \neq \mathcal{O}^i$. By introducing the eligibility vector $\mathbf{e}$, the transition and reward functions are further expanded as follows:

$$p(x_i'|\mathbf{x}, o = \mathcal{O}^i) = p(x_i'|e_i = 1)p(e_i = 1|\mathbf{x}), \tag{12}$$

$$R^i(\mathbf{x}, o = \mathcal{O}^i) = G_{\mathbf{r}}^i \mathbb{I}(e_i = 1), \tag{13}$$

where $p(x_i'|e_i = 1)$ indicates that the subtask is completed $x_i'$ if the subtask is eligible $e_i = 1$, $p(e_i|\mathbf{x})$ is the precondition $G_{\mathbf{c}}$, and $\mathbb{I}(e_i = 1)$ indicates that the reward is given only if the subtask $i$ is eligible.

## B  DETAILS OF TASK IN SUBTASK GRAPH INFERENCE PROBLEM

For self-containedness, we repeat the details of how the task (i.e., MDP) is defined by the subtask graph $G$ from (Sohn et al., 2018). We define each task as an MDP tuple $\mathcal{M}_G = (\mathcal{S}, \mathcal{A}, \mathcal{P}_G, \mathcal{R}_G, \rho_G, \gamma)$ where $\mathcal{S}$ is a set of states, $\mathcal{A}$ is a set of actions, $\mathcal{P}_G : \mathcal{S} \times \mathcal{A} \times \mathcal{S} \to [0, 1]$ is a task-specific state transition function, $\mathcal{R}_G : \mathcal{S} \times \mathcal{A} \to \mathbb{R}$ is a task-specific reward function and $\rho_G : \mathcal{S} \to [0, 1]$ is a task-specific initial distribution over states. We describe the subtask graph $G$ and each component of MDP in the following paragraphs.

**Subtask and Subtask Graph**   Formally, a subtask $\Phi^i$ is a tuple (*completion set* $\mathcal{S}_{\text{comp}}^i \subset \mathcal{S}$, *precondition* $G_{\mathbf{c}}^i$, *subtask reward* $G_{\mathbf{r}}^i \in \mathbb{R}$). The subtask $\Phi^i$ is *eligible* (i.e., subtask can be executed) if the precondition $G_{\mathbf{c}}^i$ is satisfied (see the **State Distribution and Transition Function** paragraph below for detail). The subtask $\Phi^i$ is *complete* if the current state is in the completion set $\mathbf{s}_t \in \mathcal{S}_{\text{comp}}^i$, and the agent receives the reward $\mathbf{r}_t \sim p(G_{\mathbf{r}}^i)$. Subtasks are shared across tasks. The subtask graph is a tuple of precondition and subtask reward of $N$ subtasks: $G = (G_{\mathbf{c}}, G_{\mathbf{r}})$ (see Appendix B for the detail). One example of subtask graph is given in Figure 3. A subtask graph $G$ is a tuple of the subtask reward $G_{\mathbf{r}} \in \mathbb{R}^N$, and the precondition $G_{\mathbf{c}}$ of $N$ subtasks.

**State**   The state $\mathbf{s}_t$ consists of the observation $\mathbf{obs}_t \in \{0, 1\}^{W \times H \times C}$, the completion vector $\mathbf{x}_t \in \{0, 1\}^N$, the eligibility vector $\mathbf{e}_t \in \{0, 1\}^N$, the time budget $step_t \in \mathbb{R}$ and number of episode left during the adaptation $epi_t \in \mathbb{R}$. An observation $\mathbf{obs}_t$ is represented as $H \times W \times C$ tensor, where $H$ and $W$ are the height and width of map respectively, and $C$ is the number of object types in

the domain. The $(h, w, c)$-th element of observation tensor is 1 if there is an object $c$ in $(h, w)$ on the map, and 0 otherwise. The time budget indicates the number of remaining time-steps until the episode termination. The completion vector and eligibility vector provides additional information about $N$ subtasks. The details of completion vector and eligibility vector will be explained in the following paragraph.

**State Distribution and Transition Function** Given the current state $(\mathbf{obs}_t, \mathbf{x}_t, \mathbf{e}_t)$, the next step state $(\mathbf{obs}_{t+1}, \mathbf{x}_{t+1}, \mathbf{e}_{t+1})$ is computed from the subtask graph $G$. Figure 7 describes the dependency between subtask graph and MDP. In the beginning of episode, the completion vector $\mathbf{x}_t$ is initialized to a zero vector in the beginning of the episode $\mathbf{x}_0 = [0, \dots, 0]$ and the observation $\mathbf{obs}_0$ is sampled from the task-specific initial state distribution $\rho_G$. Specifically, the observation is generated by randomly placing the agent and the $N$ objects corresponding to the $N$ subtasks defined in the subtask graph $G$. When the agent executes subtask $i$, the $i$-th element of completion vector is updated by the following update rule:

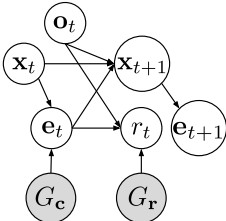

Figure 7: Dependency between subtask graph and MDP

$$x_{t+1}^i = \begin{cases} 1 & \text{if} \quad e_t^i = 1 \\ x_t^i & \text{otherwise} \end{cases}. \tag{14}$$

The observation is updated such that agent moves on to the target object, and perform corresponding primitive action. The eligibility vector $\mathbf{e}_{t+1}$ is computed from the completion vector $\mathbf{x}_{t+1}$ and precondition $G_{\mathbf{c}}$ as follows:

$$e_{t+1}^i = \underset{j \in Child_i}{\text{OR}} \left( y_{AND}^j \right), \tag{15}$$

$$y_{AND}^i = \underset{j \in Child_i}{\text{AND}} \left( \widehat{x}_{t+1}^{i,j} \right), \tag{16}$$

$$\widehat{x}_{t+1}^{i,j} = x_{t+1}^j w^{i,j} + (1 - x_{t+1}^j)(1 - w^{i,j}), \tag{17}$$

where $w^{i,j} = 0$ if there is a NOT connection between $i$-th node and $j$-th node, otherwise $w^{i,j} = 1$, and $w^{i,j}$'s are defined by the $G_{\mathbf{c}}$. Intuitively, $\widehat{x}_t^{i,j} = 1$ when $j$-th node does not violate the precondition of $i$-th node. Executing each subtask costs different amount of time depending on the map configuration. Specifically, the time cost is given as the Manhattan distance between agent location and target object location in the grid-world plus one more step for performing a primitive action.

**Task-specific Reward Function** The reward function is defined in terms of the subtask reward vector $G_{\mathbf{r}} \in \mathbb{R}^N$ and the eligibility vector $\mathbf{e}_t$, where the subtask reward vector $G_{\mathbf{r}}$ is the component of subtask graph $G$ the and eligibility vector is computed from the completion vector $\mathbf{x}_t$ and subtask graph $G$ as Eq. 17. Specifically, when agent executes subtask $i$, the amount of reward given to agent at time step $t$ is given as follows:

$$r_t = \begin{cases} G_{\mathbf{r}}^i & \text{if} \quad e_t^i = 1 \\ 0 & \text{otherwise} \end{cases}. \tag{18}$$

**Learning option** The option framework can be naturally applied to the subtask graph-based tasks. Consider the (optimal) *option* $\mathcal{O}^i = (\mathcal{I}^i, \pi_{\mathbf{o}}^i, \beta^i)$ for subtask $\Phi^i$. Its initiation set is $\mathcal{I}^i = \{\mathbf{s} | \mathbf{e}^i = 1\}$, where $\mathbf{s}$ is the state, $\mathbf{e}^i$ is the $i$-th component of eligibility vector $\mathbf{e}$, and $\mathbf{e}$ is an element of $\mathbf{s}$. The termination condition is $\beta^i = \mathbb{I}(\mathbf{x}_t^i = 1)$, where $\mathbf{x}^i$ is the $i$-th component of completion vector $\mathbf{x}$. The policy $\pi_{\mathbf{o}}^i$ maximizes the subtask reward $G_{\mathbf{r}}^i$. Similar to Andreas et al. (2017); Oh et al. (2017); Sohn et al. (2018), the option for each subtask is pre-learned via curriculum learning; *i.e.*, the agent learns options from the tasks consisting of single subtask by maximizing the subtask reward.

## C   DETAILS OF THE PLAYGROUND AND MINING DOMAIN

For self-containedness, we provide the details of Playground and Mining (Sohn et al., 2018) domain. Table 1 summarizes the complexity of the subtask graph for each task sets in Playground and Mining domain.

| | Subtask Graph Setting | | | | |
|---|---|---|---|---|---|
| | Playground | | | | Mining |
| Task | D1 | D2 | D3 | D4 | Eval |
| Depth | 4 | 4 | 5 | 6 | 4-10 |
| Subtask | 13 | 15 | 16 | 16 | 10-26 |

Table 1: (Playground) The subtask graphs in **D1** have the same graph structure as training set, but the graph was unseen. The subtask graphs in **D2**, **D3**, and **D4** have (unseen) larger graph structures. (Mining) The subtask graphs in **Eval** are unseen during training.

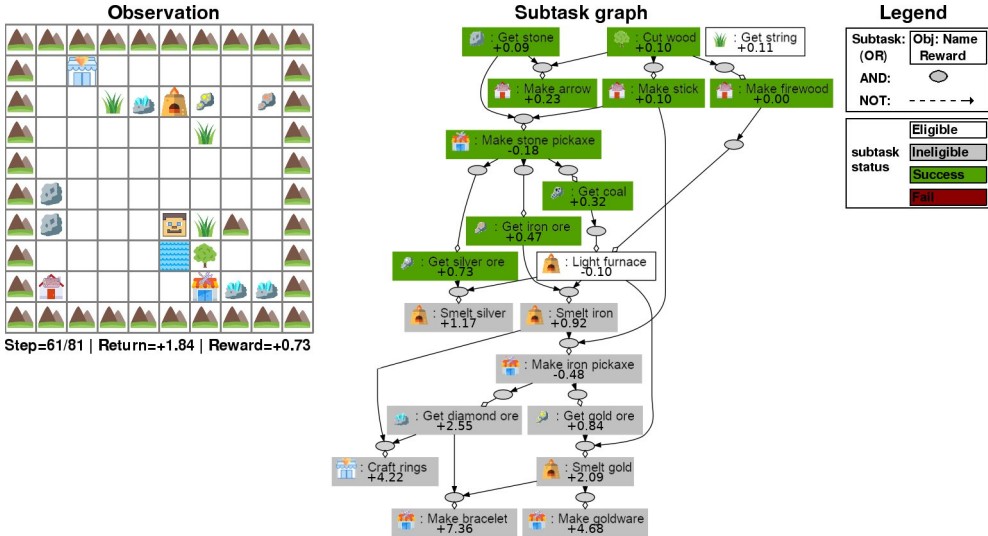

Figure 8: An example observation and subtask graph of the Mining domain (Sohn et al., 2018). The precondition of each subtask has semantic meaning based on the Minecraft game, which closely imitates the real-world tasks.

**Subtasks** The set of subtasks in **Playground** and **Mining** are implemented as $\mathcal{O} = \mathcal{A}_{int} \times \mathcal{X}$, where $\mathcal{A}_{int}$ is a set of interactions with objects, and $\mathcal{X}$ is a set of all types of interactive objects in the domain. To execute a subtask $(a_{int}, obj) \in \mathcal{A}_{int} \times \mathcal{X}$, the agent should move on to the target object $obj$ and take the primitive action $a_{int}$.

**Mining** The Mining (Sohn et al., 2018) is a domain inspired by Minecraft (see Figure 8) for example task). The agent may pickup raw materials scattered in the world. The obtained raw materials may be used to craft different items on different craft stations. There are two forms of preconditions: 1) an item may be an ingredient for building other items (e.g. stick and stone are ingredients of stone pickaxe), and 2) an item may be a required tool to pick up some objects (e.g. agent need stone pickaxe to mine iron ore). To simplify the environment, it assumes that the agent can use the item multiple times after picking it once. The subtasks in the higher layer in task graph are designed to give larger reward. The pool of subtasks and preconditions are hand-coded similar to the crafting recipes in Minecraft, and used as a template to generate 640 random task graphs. Following (Sohn et al., 2018), we used 200 for training and 440 for testing.

**Playground** The Playground (Sohn et al., 2018) domain is a more flexible domain (see Figure 3, left) which is designed to evaluate the strong generalization ability of agents on unseen task dependencies under delayed reward in a stochastic environment. More specifically, the task graph in Playground was randomly generated, hence its precondition can be any logical expression. Some of the objects randomly move, which makes the environment stochastic. The agent was trained on small

task graphs that consists of 4 layers of task dependencies, while evaluated on much larger task graphs that consists of up to 6 layers of task dependencies (See Table 1). Following (Sohn et al., 2018), we randomly generated 500 graphs for training and 500 graphs for testing. The task in the playground domain is general such that it subsumes many other hierarchical RL domains such as Taxi (Bloch, 2009), Minecraft (Oh et al., 2017) and XWORLD (Yu et al., 2017).

## D   ALGORITHM IN META-TESTING

The Algorithm 2 describes the process of our MSGI at meta-testing time for single trial.

---

**Algorithm 2** Process of single trial for a task $\mathcal{M}_G$ at meta-test time

---

**Require:** The current parameter $\theta$
**Require:** A task $\mathcal{M}_G$ parametrized by a task parameter $G$ (unknown to the agent)
1: Roll out $K$ train episodes $\tau_H = \{\mathbf{s}_t, \mathbf{o}_t, r_t, d_t\}_{t=1}^H \sim \pi_\theta^{\text{adapt}}$ in task $\mathcal{M}_G$   ▷ adaptation phase
2: Infer a subtask graph: $\widehat{G} = (\widehat{G}_{\mathbf{c}}, \widehat{G}_{\mathbf{r}}) = (\text{ILP}(\tau_H), \text{RI}(\tau_H))$   ▷ task inference
3: Roll out a test episode $\tau' = \{\mathbf{s}'_t, \mathbf{o}'_t, r'_t, d'_t\}_{t=1}^{H'} \sim \pi_{\widehat{G}}^{\text{exe}}$ in task $\mathcal{M}_G$   ▷ test phase
4: Measure the performance $R = \sum_t r'_t$ for this task

---

## E   DETAILS OF THE SC2LE DOMAIN

The **SC2LE** domain (Vinyals et al., 2017) provides suite of mini-games focusing on specific aspects of the entire StarCraft II game. In this paper, we custom design two types of new, simple mini-games called *Build Unit* and *Defeat Zerg troops*. Specifically, we built *Defeat Zerglings, Defeat Hydralisks, Defeat Hydralisks & Ultralisks* and *Build Battlecruiser* mini-games that compactly capture the most fundamental goal of the full game. The *Build Unit* mini-game requires the agent to figure-out the target unit and its precondition correctly, such that it can train the target unit within the given short time budget. The *Defeat Zerg troops* mini-game mimics the full game more closely; the agent is required to train enough units to win a war against the opponent players. To make the task more challenging and interesting, we designed the reward to be given only at the end of episode depending on the success of the whole task. Similar to the standard *Melee* game in StarCraft II, each episode is initialized with 50 *mineral*, 0 *gas*, 7 and 4 *SCV*s that start gathering mineral and gas, respectively, 1 idle *SCV*, 1 refinery, and 1 *Command Center* (See Figure 9). The episode is terminated after 2,400 environment steps (equivalent to 20 minutes in game time). In the game, the agent is initially given 50 *mineral*, 0 *gas*, 7 and 4 *SCV*s that start gathering mineral and gas, respectively, 1 idle *SCV*, 1 refinery, and 1 *Command Center* (See Figure 9) and is allowed to prepare for the upcoming battle only for 2,400 environment steps (equivalent to 20 minutes in game time). Therefore, the agent must learn to collect resources and efficiently use them to build structures for training units. All the four custom mini-games share the same initial setup as specified in Figure 9.

*Defeat Zerg troops* **scenario**: At the end of the war preparation, different combinations of enemy unit appears: *Defeat Zerglings* and *Defeat Hydralisks* has 20 zerglings and 15 hydralisks, respectively, and *Defeat Hydralisks & Ultralisks* contains a combination of total 5 hydralisks and 3 ultralisks. When the war finally breaks out, the units trained by the agent will encounter the army of Zerg units in the map and combat until the time over (240 environment steps or 2 minutes in the game) or either side is defeated. Specifically, the agent may not take any action, and the units trained by the agent perform an *auto attack* against the enemy units. Unlike the original full game that has ternary reward structure of +1 (win) / 0 (draw) / −1 (loss), we use binary reward structure of +1 (win) and −1 (loss or draw). Notice that depending on the type of units the agent trained, a draw can happen. For instance, if the units trained by the agent are air units that cannot attack the ground units and the enemy units are the ground units that cannot attack the air units, then no combat will take place, so we consider this case as a loss. *Build unit* **scenario**: The agent receives the reward of +1 if the target unit is successfully trained within the time limit, and the episode terminates. When the episode terminates due to time limit, the agent receives the reward of −1. We gave 2,400 step budget for the *Build Battlecruiser* scenario such that only highly efficient policy can finish the task within the time limit.

The transition dynamics (*i.e.*, build tech-tree) in **SC2LE** domain has a hierarchical characteristic which can be inferred by our MSGI agent (see Figure 9). We conducted the experiment on Terran race only, but our method can be applied to other races as well.

**Subtask.** There are 85 subtasks: 15 subtasks of constructing each type of building (*Supply depot, Barracks, Engineeringbay, Refinery, Factory, Missile turret, Sensor tower, Bunker, Ghost academy, Armory, Starport, Fusioncore, Barrack-techlab, Factory-techlab, Starport-techlab*), 17 subtasks of training each type of unit (*SCV, Marine, Reaper, Marauder, Ghost, Widowmine, Hellion, Hellbat, Cyclone, Siegetank, Thor, Banshee, Liberator, Medivac, Viking, Raven, Battlecruiser*), one subtask of idle worker, 32 subtasks of selecting each type of building and unit, gathering mineral, gathering gas, and no-op. For gathering mineral, we set the subtask as (mineral$\geq val$) where $val \in \{50, 75, 100, 125, 150, 300, 400\}$. Similarly for gathering gas, we set the subtask as (gas$\geq val$) where $val \in \{25, 50, 75, 100, 125, 150, 200, 300\}$. For no-op subtask, the agent takes the no-op action for 8 times.

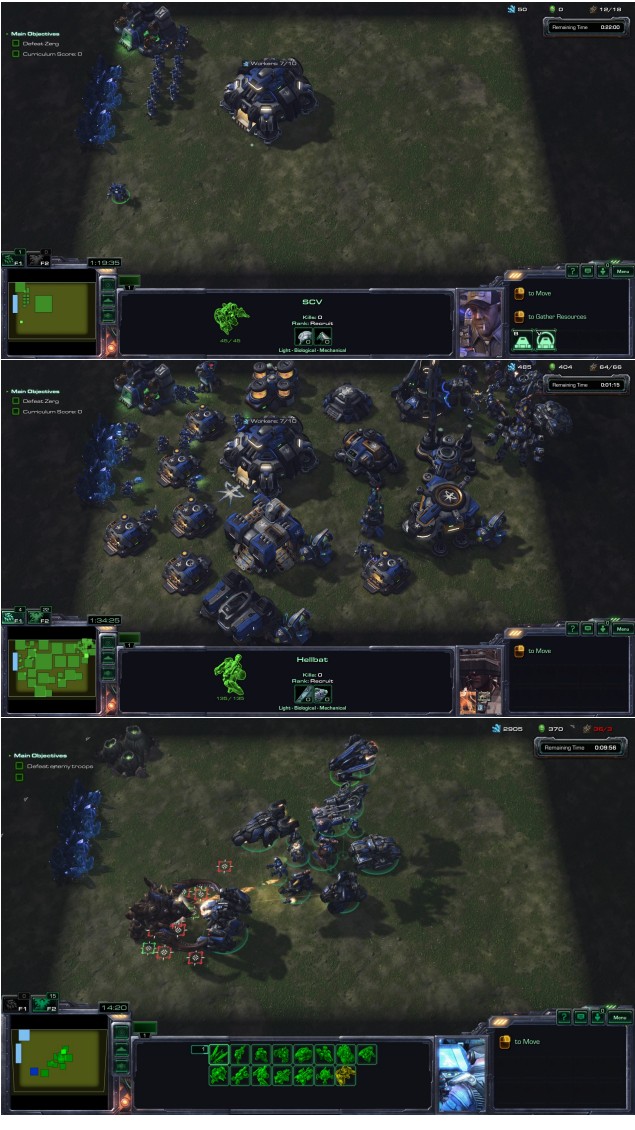

Figure 9: (**Top**) The agent starts the game initially with limited resources of 50 minerals, 0 gases, 3 foods, 11 SCVs collecting resources, 1 idle SCV and pre-built Refinery. (**Middle**) From the initial state, the agent needs to strategically collect resources and build structures in order to be well prepared for the upcoming battle. (**Bottom**) After 2,400 environment steps, the war breaks; all the buildings in the map are removed, and the enemy units appear. The agent's units should eliminate the enemy units within 240 environment steps during the war.

**Eligibility.** The eligibility of the 15 building construction subtasks and 17 training unit subtasks is given by the environment as an *available action* input. For the selection subtasks, we extracted the number of corresponding units using the provided API of the environment. Gathering mineral, gas, and no-op subtasks are always eligible.

**Completion.** The completion of the 15 construction subtasks and 17 training subtasks is 1 if the corresponding building or unit is present on the map. For the selection subtasks, the completion is 1 if the target building or unit is selected. For gathering mineral and gas subtasks, the subtask is completed if the condition is satisfied (*i.e.*, gas$\geq 50$). The no-op subtask is never completed.

**Subtask reward.** In SC2LE domain, the agent does not receive any reward when completing a subtask. The only reward given to agent is the binary reward $r_{H_{\mathrm{epi}}} = \{+1, -1\}$ at the end of episode (*i.e.*, $t = H_{\mathrm{epi}}$). Therefore, the subtask reward inference method described in Eq.(4) may not be applied. Instead, we tried to infer the subtask reward $\widehat{G}_{\mathbf{r}} \in \mathbb{R}^N$ (see Section 3 for definition) from a victory reward $r_{H_{\mathrm{epi}}}$ by building a binary classifier that predicts the victory reward $r_{H_{\mathrm{epi}}}$ from the option count vector $\mathbf{n} \in \mathbb{N}^N$ using a logistic model (*i.e.*, logistic regression), where $N$ is the number of subtasks and the option count vector $\mathbf{n}$ counts how many times each option had been executed within an episode. Intuitively speaking, we assume that the execution of each subtask (*i.e.*, option) gives an implicit subtask reward that is un-observable by the agent, and the victory reward is determined by thresholding the sum of subtask rewards within an episode as follows:

$$r_{H_{\mathrm{epi}}} = \mathbb{I}(G_{\mathbf{r}}^{\top}\mathbf{n} > \beta), \tag{19}$$

where $\mathbb{I}(\cdot)$ is the indicator function and $\beta$ is the threshold. Then, we approximate it using a sigmoid function $\sigma(\cdot)$ as follows:

$$r_{H_{\mathrm{epi}}} = \sigma(G_{\mathbf{r}}^{\top}\mathbf{n} - \beta). \tag{20}$$

In the adaptation phase, we randomly sampled the subtask reward vector $G_{\mathbf{r}}$ from the uniform distribution in $[0, 1]^N$, and used it for running MSGI-GRProp agent while recording the option count vector $\mathbf{n}$. Then, the option count vectors $\mathbf{n}$ and the victory rewards $r_{H_{\mathrm{epi}}}$ from the $K$ episodes in adaptation phase form a training data $\left\{\mathbf{n}_i, r_{H_{\mathrm{epi}},i}\right\}_{i=1}^{K}$ for estimating the parameters of logistic model as follows:

$$\log \frac{r_{H_{\mathrm{epi}}}}{1 - r_{H_{\mathrm{epi}}}} = \widehat{G}_{\mathbf{r}}^{\top}\mathbf{n} - \beta, \tag{21}$$

where $\widehat{G}_{\mathbf{r}} \in \mathbb{R}^N$ and $\beta \in \mathbb{R}$ are the weight and bias parameters to be learned. Finally, we used the learned $\widehat{G}_{\mathbf{r}}$ as the subtask reward vector for running our MSGI-GRProp agent in test phase. We used the *scikit-learn* (Pedregosa et al., 2011) implementation of logistic regression.

## F  MORE RESULTS ON THE SC2LE DOMAIN

**Accuracy of inferred subtask graph.** Figure 10 shows the accuracy of the subtask graph inferred by MSGI-GRProp agent (Section 5.2), in terms of precision and recall over different adaption horizon.

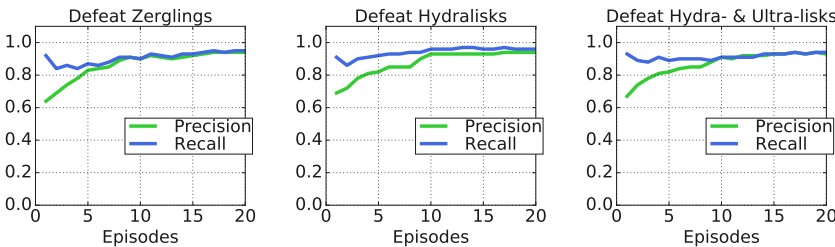

Figure 10: Precision and recall of binary assignments on the inferred subtask graph's precondition.

**Qualitative Examples.** Figure 12 shows a simplified form of the subtask graph inferred by our MSGI-GRProp agent after 20 episodes of adaptation. For better readability, we removed the preconditions of resources (food, mineral, gas); Figure 13 depicts the full subtask graph. Compared to the actual tech-tree of the game, we can see the dependency between buildings and units are correctly inferred.

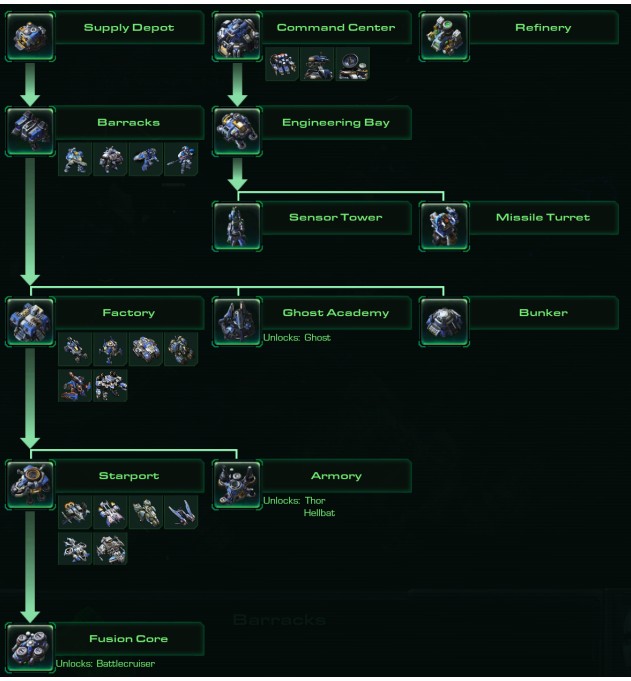

Figure 11: The actual tech-tree of Terran race in StarCraft II. There exists a hierarchy in the task, which can be autonomously discovered by our MSGI agent.

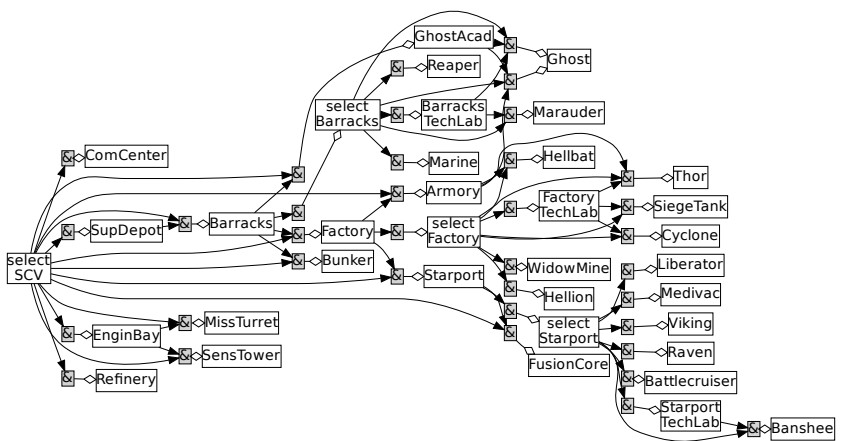

Figure 12: A simplified version of subtask graph inferred by our MSGI-GRProp agent after 10 episodes of adaptation.

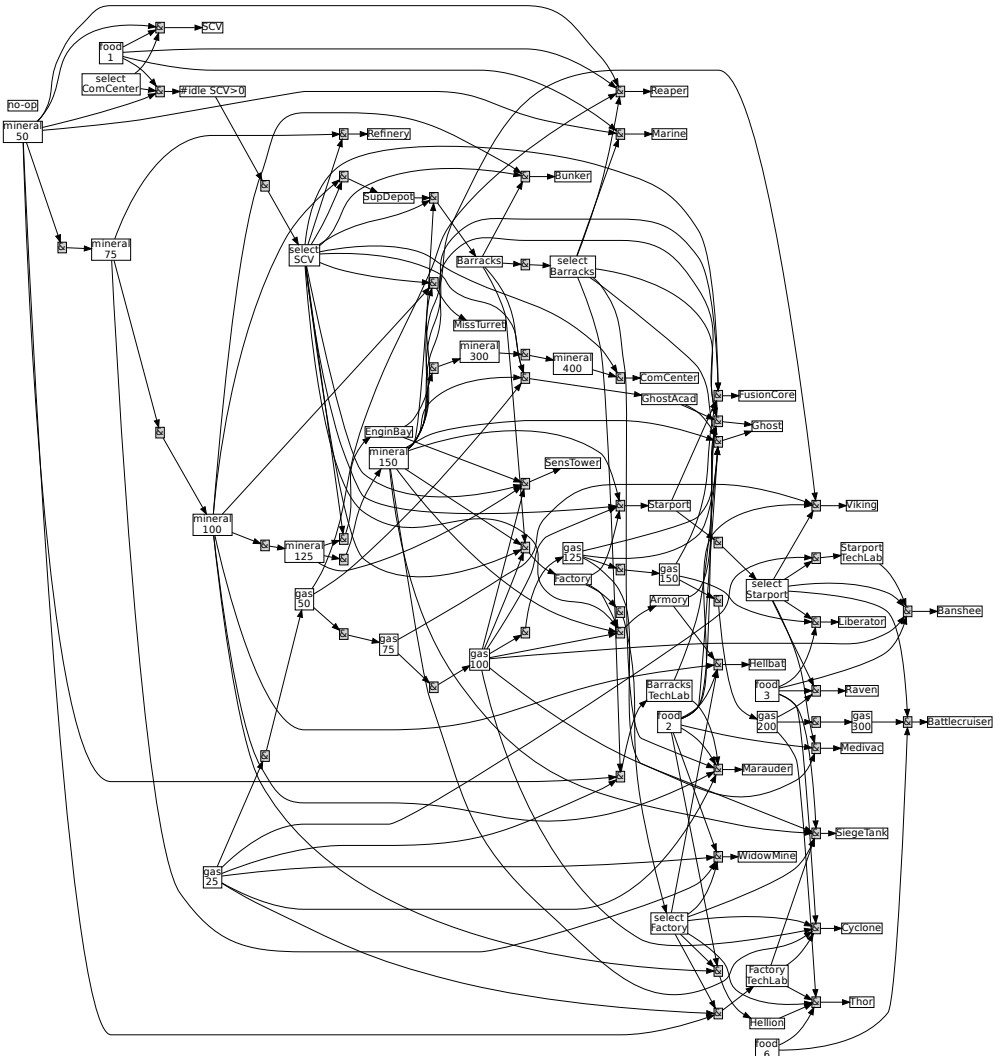

Figure 13: The full subtask graph inferred by our MSGI agent.

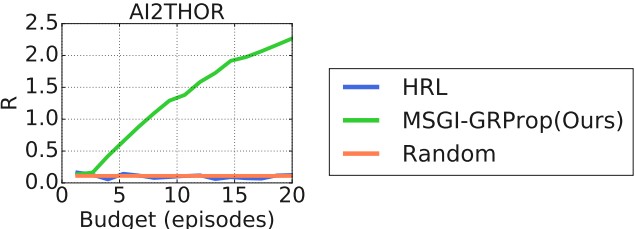

Figure 14: Adaptation performance of MSGI-GRProp, Random, and HRL agents with different adaptation horizon on **AI2-THOR** domain. The episode terminates after the agent executes 20 subtasks or when there is no subtask available to execute. Our MSGI-GRProp achieves around 2.5 total reward within an episode by executing roughly two *serve* subtasks, while the baseline methods almost never get any reward.

## G   EXPERIMENT ON AI2-THOR DOMAIN

The **AI2-THOR** (Kolve et al., 2017) is an interactive 3D environment where the agent can both navigate and interact with the objects within the environment through variety of actions that can change the states of the object (*i.e.*, the *PickupObject* action changes the object's *isPickedUp* state). Among the several scenes provided by the environment, we focus on the kitchen scene and evaluate agents on the task similar to the *breakfast preparation* task described in the introduction: The agent is required to prepare the dishes by directly manipulating the objects given in the scene. See Figure 15 for an example task.

There are two different types of objects in the scene: The first type is the plain object (*i.e.*, *Bread, Apple, Potato, Tomato, Egg, Lettuce, Cup, Mug, Plate, Pan, Bowl*) that agents can move around with, and the second type is the receptacle object (*i.e.*, *Pan, Plate, Bowl, Cabinet, Microwave, StoveBurner, CounterTop, DiningTable, SideTable, Toaster, CoffeeMachine, Fridge*) which can contain other objects depending on their sizes. With these objects and the subtasks defined, the agent is required to cook and serve foods through long sequence of subtasks. For instance, the agent can prepare a *fried egg* dish by (1) placing a *Pan* object on *StoveBurner*, (2) placing *Egg* on *Pan*, (3) slicing (or, cracking) *Egg* to *EggCracked*, (4) turning on the *StoveKnob* to cook the cracked egg, and finally (5) serving the cooked *Egg* on the *Plate*. Rewards are given only when the agent successfully serves the cooked object on the appropriate receptacles. Similar to the **SC2LE** domain (see Section 5.2), the task structure in **AI2-THOR** is fixed as well (*i.e.*, *Egg* cannot be cooked in the *Fridge*), and thus instead of training meta agents, we evaluate and compare MSGI-GRProp, HRL, and Random agents on the cooking tasks. The visualization of the subtask graph (*i.e.*, underlying task structure) *inferred* by MSGI-GRProp agent is available on the Figure 16.

**Subtask.** There are total 148 subtasks: 17 subtasks for picking up all the possible objects in the scene (*Tomato, Potato, Lettuce, Apple, Egg, Bread, TomatoSliced, PotatoSliced, LettuceSliced, AppleSliced, EggCracked, BreadSliced, Pan, Plate, Cup, Mug, Bowl*), 113 subtasks for putting down each pickupable objects into pre-defined putdownable receptacles, 6 subtasks for slicing the sliceable objects, 6 subtasks for cooking cookable objects or filling up liquid in the *Mug* or *Cup*, and 6 subtasks for serving the cooked or filled objects on the proper receptacles such as *Plate*, *Bowl*, and *DiningTable*.

**Completion.** The completion of the 17 pick up subtasks is 1 if the corresponding object is in the agent's hand or inventory. For all the put down subtasks, the completion is 1 if the corresponding object is in the target receptacle (*i.e.*, *TomatoSliced* on *Plate*). For slice and cook subtasks, the completion is 1 if the corresponding object is sliced or cooked, respectively. The serve subtasks are complete if the corresponding objects are placed on the target receptacles (*i.e.*, *Cooked PotatoSliced* on *Plate*). When the agent executes the *serve* subtask, the served food is removed to simulate the user eating the served dish, such that the agent can execute the same *serve* subtask at most once within the episode.

**Eligibility.** The eligibility of the subtasks is computed based the corresponding subtask completion vector. The eligibility of the pickup subtasks is always set to 1, and the putdown subtasks are eligible if the corresponding object is picked up (*i.e.*, *Pickup Tomato* is complete). The slice subtasks are eligible if the sliceable objects are on any receptacle. The cook subtasks are eligible if the cookable

objects are placed on the corresponding cooking station (*i.e.*, *Mug* is on the *CoffeeMachine*). The eligibility of the serve subtasks is 1 if the corresponding objects are either cooked or sliced.

**Subtask reward.** To make the task more challenging and realistic, we assigned a non-zero reward only to the six *Serve* subtasks, that have the most complex precondition (*i.e.*, sparse reward setting). Similar to other environments, we randomly set the subtask reward of each subtask from the predefined range when we sample a new task (*i.e.*, trial). Table 2 specifies the range of subtask reward for the six *serve* subtasks; intuitively speaking, we set a higher subtask reward for the subtask that has more complex precondition.

| Subtask name | Min reward | Max reward |
|---:|:---:|:---:|
| *Serve cooked potato* | 0.6 | 1.2 |
| *Serve cooked and sliced potato* | 1.0 | 2.0 |
| *Serve cooked and sliced bread* | 1.0 | 2.0 |
| *Serve cooked and cracked egg* | 1.0 | 2.0 |
| *Serve coffee* | 0.6 | 1.2 |
| *Serve water* | 0.4 | 1.0 |

Table 2: The range from which the subtask reward of *serve* subtask was sampled, in the **AI2-THOR** environment.

**Result.** On the **AI2-THOR** environment, we compared our MSGI-GRProp agent with two baseline agents: Random and HRL. Figure 14 summarizes the performance of each agent with varying adaptation budgets. We observed that both the Random and HRL agent almost never receives any non-zero reward during 20 episodes of adaptation, since the *serve* subtasks have a complex precondition that is not easy to satisfy for random policy. In contrast, our MSGI-GRProp agent achieves around 2.4 total reward on average after 20 episodes of adaptation. As specified in Table 2, each *serve* subtask gives around 1.0~1.5 reward, so it means MSGI-GRProp agent executes around two *serve* subtasks within an episode. Also, considering that the minimum number of subtasks required for *serve* subtask is around 6, being able to execute around two subtasks within only 20 steps means the agent does not waste its time for executing other subtasks that are irrelevant to the target *serve* subtasks. The HRL agent's performance does not improve during adaptation since it seldom observes any reward. On the other hand, our MSGI-GRProp agent can quickly find a way to execute the *serve* subtasks by inferring the precondition of them; as shown in Figure 16, our ILP module can accurately and efficiently infer the precondition of subtasks in **AI2-THOR** environment after only 20 episodes of adaptation.

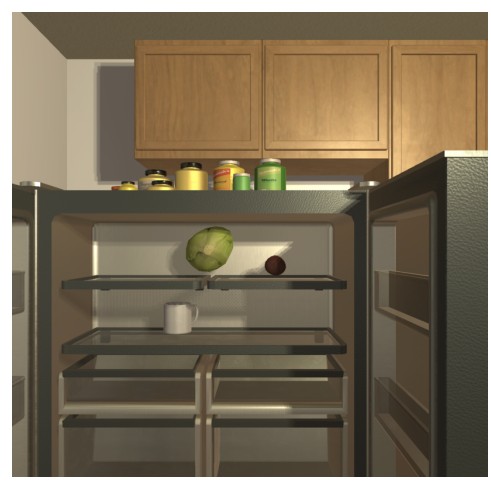

(a) Pick up *Potato* from *Fridge*.

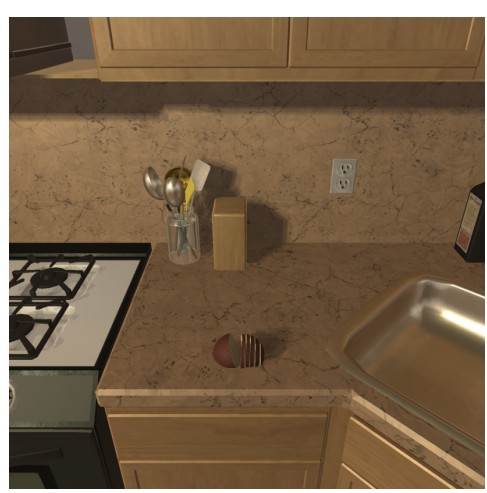

(b) Slice *Potato* to *PotatoSliced*.

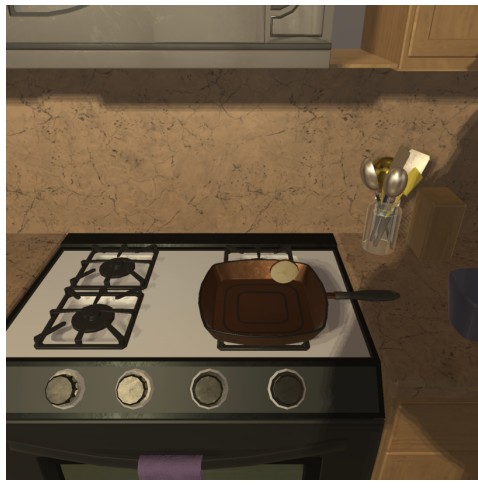

(c) Place *PotatoSliced* on *Pan* and cook.

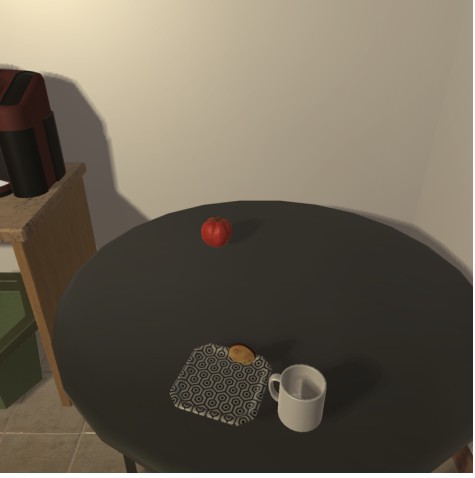

(d) Serve *Cooked PotatoSliced* on *Plate*.

Figure 15: (a) - (d) demonstrates an example task of preparing fried potato in the **AI2-THOR** domain. The *serve PotatoSliced on Plate* subtask requires slicing the potato (*e.g.*, (b)) and frying the sliced potato on the pan (*e.g.*, (c)) before serving the dish. The agent receives a reward after finishing the final subtask (*e.g.*, (d)) of serving the dish on the plate.

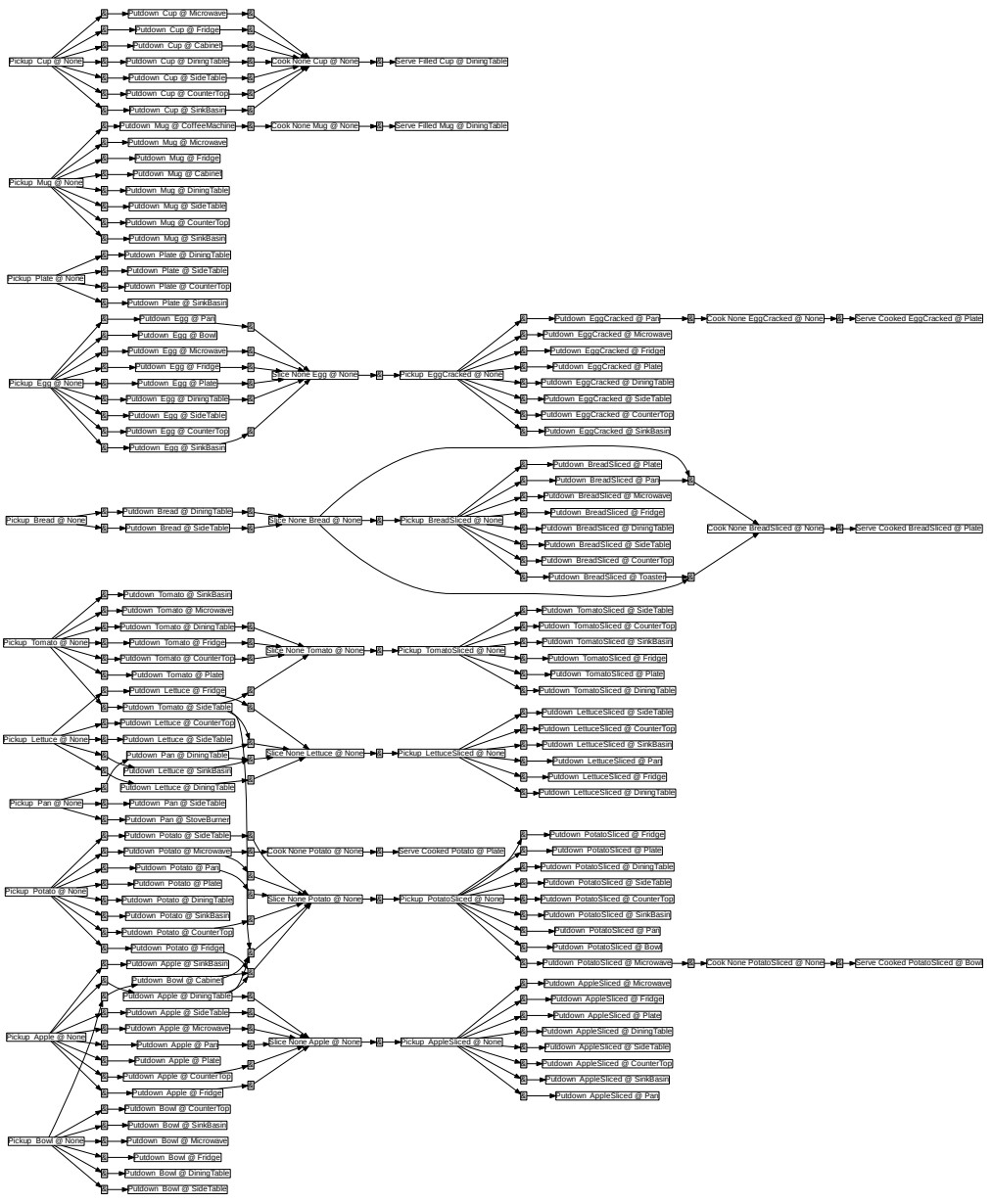

Figure 16: The subtask graph inferred by our MSGI-GRProp agent in the **AI2-THOR** environment after 20 episodes.

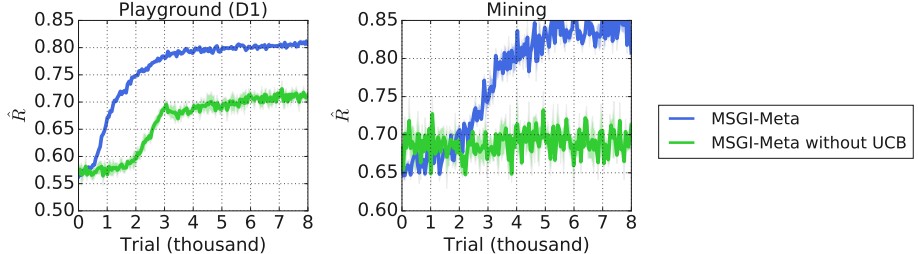

Figure 17: Comparison of meta-training MSGI-Meta agent that was trained with UCB bonus and extrinsic reward, and MSGI-Meta without UCB agent that was trained with extrinsic reward only in the **Playground** and **Mining** domain. In both domains, adding UCB bonus improves the meta-training performance of our MSGI-Meta agent.

# H MORE RESULTS ON MINING AND PLAYGROUND

## H.1 ABLATION STUDY ON THE INTRINSIC REWARD

We conducted an ablation study comparing our MSGI-Meta with and without UCB bonus. We will refer our method with UCB bonus as MSGI-Meta, and our method without UCB bonus as MSGI-Meta without UCB. Figure 17 shows that UCB bonus facilitates the meta-training of our MSGI-Meta agents in both **Playground** and **Mining** domains.

## H.2 QUALITATIVE RESULT ON THE SUBTASK GRAPH INFERENCE

Figure 18 illustrates a qualitative example of the inferred subtask graphs inferred by MSGI-Meta and MSGI-Rand agents on the **Mining**-Eval set. The adaptation budget was $K = 50$ episodes and episode length was $T = 80$ steps. Both of MSGI-Meta and MSGI-Rand correctly inferred most of subtasks in the lower hierarchy (*e.g.*, *Get stone*, *Cut wood*, *Get string*) of the subtask graph. However, only MSGI-Meta was successful in inferring the preconditions of subtasks in the highest hierarchy (*e.g.*, *Smelt gold*, *Make goldware*, and *Craft necklace*); MSGI-Rand never had an experience where their preconditions are all satisfied, and thus failed to learn the preconditions of these task. It demonstrates that MSGI-Meta with a meta-learned adaptation policy is able to collect more comprehensive experience for accurate subtask graph inference.

## H.3 QUANTITATIVE ANALYSIS OF THE ADAPTATION POLICY

We measured the portion of subtasks being eligible or completed at least once (i.e., coverage) during adaptation to measure how exploratory MSGI-Meta and random policy are. We report the averaged coverage over

| Method | Coverage (%) | | | | |
|---|---|---|---|---|---|
| | **D1** | **D2** | **D3** | **D4** | **Eval** |
| MSGI-Meta | **89** | **87** | **81** | **75** | **90** |
| MSGI-Rand | 83 | 77 | 68 | 58 | 85 |

the evaluation graph set and 8 random seeds. The table shows that MSGI-Meta can make more diverse subtasks complete and eligible than the random policy thanks to more accurate subtask graph inference.

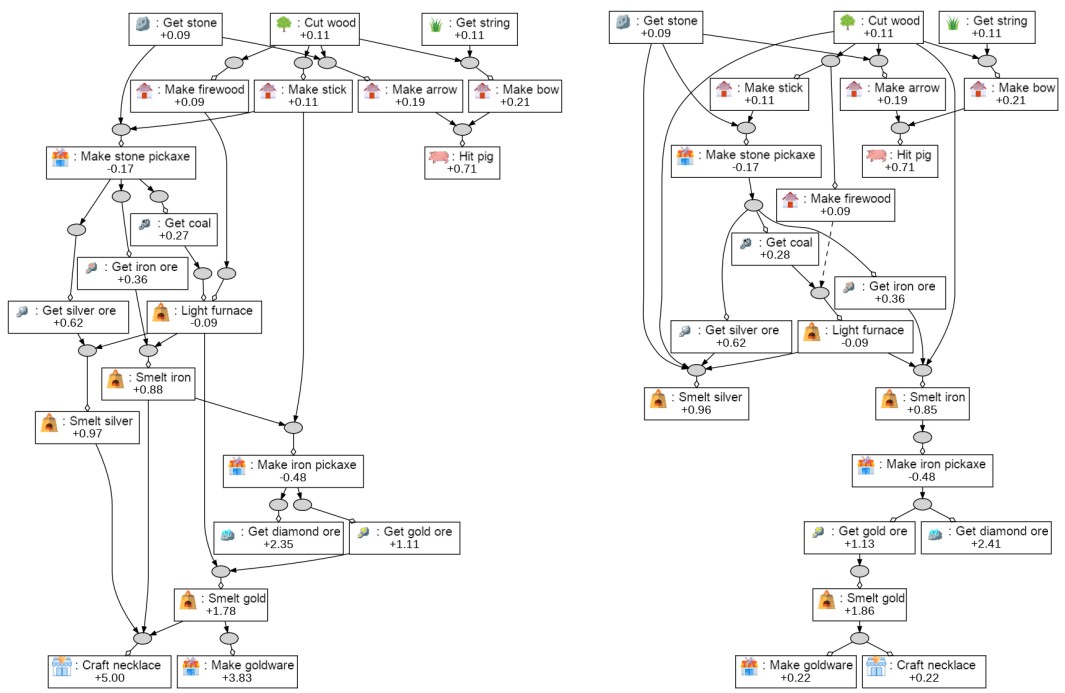

(a) A ground-truth subtask graph.     (b) A subtask graph inferred by MSGI-Meta.

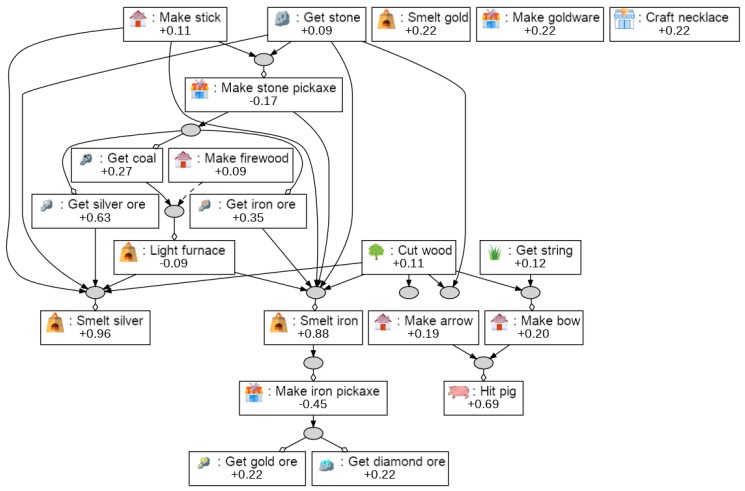

(c) A subtask graph inferred by MSGI-Rand.

Figure 18: A qualitative example of subtask graph inference, in the **Mining** domain.

# I    DETAILS OF GRPROP POLICY

For self-containedness, we provide the description of GRProp policy from Sohn et al. (2018). We also make a few modifications on $\widetilde{\text{OR}}(\mathbf{x})$ and $\widetilde{\text{AND}}(\mathbf{x})$ in Eqs. 29 and 30.

Intuitively, GRProp policy modifies the subtask graph to a differentiable form such that we can compute the gradient of modified return with respect to the subtask completion vector in order to measure how much each subtask is likely to increase the modified return. Let $\mathbf{x}_t$ be a completion vector and $G_{\mathbf{r}}$ be a subtask reward vector (see Section 2 for definitions). Then, the sum of reward until time-step $t$ is given as:

$$U_t = G_{\mathbf{r}}^{\top}\mathbf{x}_t. \tag{22}$$

We first modify the reward formulation such that it gives a half of subtask reward for satisfying the preconditions and the rest for executing the subtask to encourage the agent to satisfy the precondition of a subtask with a large reward:

$$\widehat{U}_t = G_{\mathbf{r}}^{\top}(\mathbf{x}_t + \mathbf{e}_t)/2. \tag{23}$$

Let $y_{AND}^j$ be the output of $j$-th AND node. The eligibility vector $\mathbf{e}_t$ can be computed from the subtask graph $G$ and $\mathbf{x}_t$ as follows:

$$e_t^i = \underset{j \in Child_i}{\text{OR}}\left(y_{AND}^j\right), \quad y_{AND}^j = \underset{k \in Child_j}{\text{AND}}\left(\widehat{x}_t^{j,k}\right), \quad \widehat{x}_t^{j,k} = x_t^k w^{j,k} + \text{NOT}(x_t^k)(1 - w^{j,k}), \tag{24}$$

where $w^{j,k} = 0$ if there is a NOT connection between $j$-th node and $k$-th node, otherwise $w^{j,k} = 1$. Intuitively, $\widehat{x}_t^{j,k} = 1$ when $k$-th node does not violate the precondition of $j$-th node. The logical AND, OR, and NOT operations in Eq. 24 are substituted by the smoothed counterparts as follows:

$$p^i = \lambda_{\text{or}}\widetilde{e}^i + (1 - \lambda_{\text{or}})\,x^i, \tag{25}$$

$$\widetilde{e}^i = \underset{j \in Child_i}{\widetilde{\text{OR}}}\left(\widetilde{y}_{AND}^j\right), \tag{26}$$

$$\widetilde{y}_{AND}^j = \underset{k \in Child_j}{\widetilde{\text{AND}}}\left(\hat{x}^{j,k}\right), \tag{27}$$

$$\hat{x}^{j,k} = w^{j,k}p^k + (1 - w^{j,k})\widetilde{\text{NOT}}\left(p^k\right), \tag{28}$$

where $\mathbf{x} \in \mathbb{R}^d$ is the input completion vector,

$$\widetilde{\text{OR}}(\mathbf{x}) = \text{softmax}(w_{\text{or}}\mathbf{x}) \cdot \mathbf{x}, \tag{29}$$

$$\widetilde{\text{AND}}(\mathbf{x}) = \frac{\zeta(\mathbf{x}, w_{\text{and}})}{\zeta(||\mathbf{x}||, w_{\text{and}})}, \tag{30}$$

$$\widetilde{\text{NOT}}(\mathbf{x}) = -w_{\text{not}}\mathbf{x}, \tag{31}$$

$||\mathbf{x}|| = d$, $\zeta(\mathbf{x}, \beta) = \frac{1}{\beta}\log(1 + \exp(\beta\mathbf{x}))$ is a soft-plus function, and $\lambda_{\text{or}} = 0.6, w_{\text{or}} = 2, w_{\text{and}} = 3, w_{\text{not}} = 2$ are the hyper-parameters of GRProp. Note that we slightly modified the implementation of $\widetilde{\text{OR}}$ and $\widetilde{\text{AND}}$ from sigmoid and hyper-tangent functions in (Sohn et al., 2018) to softmax and softplus functions for better performance. With the smoothed operations, the sum of smoothed and modified reward is given as:

$$\widetilde{U}_t = G_{\mathbf{r}}^{\top}\mathbf{p}, \tag{32}$$

where $\mathbf{p} = [p^1, \ldots, p^d]$ and $p^i$ is computed from Eq. 25. Finally, the graph reward propagation policy is a softmax policy,

$$\pi(\mathbf{o}_t|G, \mathbf{x}_t) = \text{Softmax}\left(T\nabla_{\mathbf{x}_t}\widetilde{U}_t\right) = \text{Softmax}\left(TG_{\mathbf{r}}^{\top}(\lambda_{\text{or}}\nabla_{\mathbf{x}_t}\widetilde{\mathbf{e}}_t + (1 - \lambda_{\text{or}}))\right), \tag{33}$$

where we used the softmax temperature $T = 40$ for **Playground** and **Mining** domain, and linearly annealed the temperature from $T = 1$ to $T = 40$ during adaptation phase for **SC2LE** domain. Intuitively speaking, we act more confidently (*i.e.*, higher temperature $T$) as we collect more data since the inferred subtask graph will become more accurate.

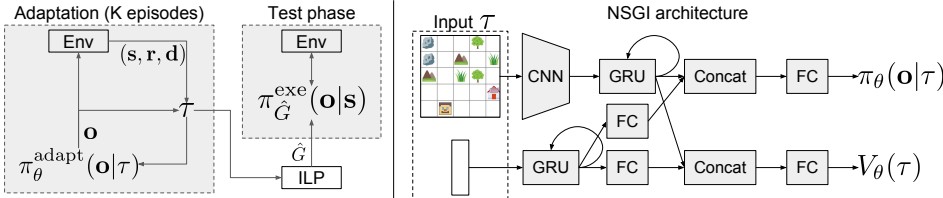

Figure 19: (Left) Our MSGI model and (Right) the architecture of adaptation policy $\pi_\theta^{\text{adapt}}$.

## J    Implementation Details

### J.1    Details of MSGI architecture

Figure 19 illustrates the architecture of our MSGI model. Our adaptation policy takes the agent's trajectory $\tau_t = \{\mathbf{s}_t, \mathbf{o}_t, \mathbf{r}_t, \mathbf{d}_t\}$ at time step $t$ as input, where $\mathbf{s} = \{\text{obs}, \mathbf{x}, \mathbf{e}, \text{step}, \text{epi}\}$. We used convolutional neural network (CNN) and gated rectifier unit (GRU) to encode both the temporal and spatial information of observation input obs. For other inputs, we simply concatenated all of them along the dimension after normalization, and encoded with GRU and fully-connected (FC) layers. Finally, the flat embedding and observation embedding are concatenated with separate heads for the value and policy output respectively (See supplemental material for more detail).

Our MSGI architecture encodes the observation input using CNN module. Specifically, the observation embedding is computed by Conv1(16x1x1-1/0)-Conv2(32x3x3-1/0)-Conv3(64x3x3-1/1)-Conv4(32x3x3-1/1)-Flatten-FC(512)-GRU(512). Other inputs are all concatenated into a single vector, and fed to GRU(512). In turn, we extracted two flat embeddings using two separate FC(512) heads for policy and value outputs. For each output, the observation and flat embeddings and concatenated into single vector, and fed to FC(512)-FC($d$) for policy output and FC(512)-FC(1) for value output, where $d$ is the policy dimension. We used ReLU activation function in all the layers.

### J.2    Details of Training MSGI-Meta

Algorithm 1 describes the pseudo-code for training our MSGI-Meta model with and without UCB bonus term. In adaptation phase, we ran a batch of 48 parallel environments. In test phase, we measured the average performance over 4 episodes with 8 parallel workers (*i.e.*, average over 32 episodes). We used actor-critic method with GAE (Schulman et al., 2016) as follows:

$$\nabla_\theta \mathcal{L} = \mathbb{E}_{G \sim \mathcal{G}_{train}} \left[ \mathbb{E}_{s \sim \pi_\theta} \left[ -\nabla_\theta \log \pi_\theta \sum_{l=0}^{\infty} \left( \prod_{n=0}^{l-1} (\gamma\lambda)^{k_n} \right) \delta_{t+l} \right] \right], \tag{34}$$

$$\delta_t = r_t + \gamma^{k_t} V_\theta^\pi(\mathbf{s}_{t+1}) - V_\theta^\pi(\mathbf{s}_t), \tag{35}$$

where we used the learning rate $\eta = 0.002$, $\gamma = 1$, and $\lambda = 0.9$. We used RMSProp optimizer with the smoothing parameter of 0.99 and epsilon of 1e-5. We trained our MSGI-Meta agent for 8000 trials, where the agent is updated after every trial. We used the best hyper-parameters chosen from the sets specified in Table 4 for all the agents. We also used the entropy regularization with annealed parameter $\beta_{\text{ent}}$. We started from $\beta_{\text{ent}} = 0.05$ and linearly decreased it after 1200 trials until it reaches $\beta_{\text{ent}} = 0$ at 3200 trials. During training, we update the critic network to minimize $\mathbb{E}[(R_t - V_\theta^\pi(\mathbf{s}_t))^2]$, where $R_t$ is the cumulative reward at time $t$ with the weight of 0.03. We clipped the magnitude of gradient to be no larger than 1.

### J.3    Details of training $RL^2$ and HRL

For training $RL^2$ and HRL, we used the same architecture and algorithm with MSGI-Meta. For $RL^2$, we used the same hyper-parameters except the learning rate $\eta = 0.001$ and the critic loss weight of 0.005. For HRL, we used the learning rate $\eta = 0.001$ and the critic loss weight of 0.12. We used the best hyper-parameters chosen from the sets specified in Table 4 for all the agents.

| Hyperparameter | Notation | Methods | | |
|---|---|---|---|---|
| | | MSGI-Meta | RL$^2$ | HRL |
| Learning Rate (LR) | $\eta$ | 2e-3 | 1e-3 | 1e-3 |
| LR multiplier | | 0.999 | 0.999 | 0.999 |
| GAE | $\lambda$ | 0.9 | 0.9 | 0.9 |
| Critic | $\beta_{\text{critic}}$ | 0.12 | 0.005 | 0.12 |
| Entropy | $\beta_{\text{ent}}$ | 0.1 | 1.0 | 0.03 |
| UCB | $\beta_{\text{UCB}}$ | 1.0 | - | - |
| Architecture | $(d_{\text{flat}}, d_{\text{gru}})$ | (512, 512) | (512, 512) | (512, 512) |

Table 3: Summary of hyper-parameters used for MSGI-Meta, RL$^2$, and HRL agents.

| Hyperparameter | Notation | Values |
|---|---|---|
| Learning rate (LR) | $\eta$ | {1.0, 2.5, 5.0}×{e-5, e-4, e-3} |
| LR multiplier | | {0.96, 0.98, 0.99, 0.993, 0.996, 0.999, 1.0} |
| GAE | $\lambda$ | {0.1, 0.2, 0.3, 0.4, 0.5, 0.6, 0.7, 0.8, 0.9, 0.95, 0.98, 1.0} |
| Critic | $\beta_{\text{critic}}$ | {0.005, 0.01, 0.03, 0.06, 0.12, 0.25} |
| Entropy | $\beta_{\text{ent}}$ | {0.02, 0.05, 0.1, 0.2, 0.5, 1.0, 2.0} |
| UCB | $\beta_{\text{UCB}}$ | {1.0, 3.0} |
| Architecture | $(d_{\text{flat}}, d_{\text{gru}})$ | {(128, 128), (256, 256), (512, 512)} |

Table 4: The range of hyper-parameters we searched over. We did beam-search to find the best parameter with the priority of $\eta, \lambda, \beta, \beta_{\text{ent}}, (d_{\text{flat}}, d_{\text{gru}})$, LR-decay.

