# OpenReview forum: "Meta Reinforcement Learning with Autonomous Inference of Subtask Dependencies"
_ICLR.cc/2020/Conference — Accept (Poster)_

### Official Review · AnonReviewer3 · 2019-10-19
**Official Blind Review #564**

**Rating:** 6

**Review:**

The main problem that is tackled here are tasks that have a main goal that can only be reached by solving prerequisite tasks. They test their method on a simple game and a very complex one.

Methodology and novelty
The authors combine various techniques (subtask graph inference, gradient based meta-learning and inductive logic programming). It is not clearly stated if the authors combined techniques and/or if they invented a new one. What is the big difference from the work by Sohn et al. (2018)?

Experiments
The authors evaluated one agent. It would have been better if they trained multiple agents and showed a performance distribution, so it is clear that the performance is not just achieved by luck (Fig 5.).
The video material showed clearly how the complex game (StarCraft II) was solved much quicker than a baseline model.

Presentation
Figure 3 does not give a description of the subtask graph (middle) and the StarCraft II. The video material clearly shows the performance of their method. Section 5.1.2 does not clearly explain the different datasets D1-D5 of Playground.

**Experience Assessment:**

I do not know much about this area.

**Review Assessment: Checking Correctness Of Derivations And Theory:**

I did not assess the derivations or theory.

**Review Assessment: Checking Correctness Of Experiments:**

I did not assess the experiments.

**Review Assessment: Thoroughness In Paper Reading:**

I made a quick assessment of this paper.

---

> ### Author Response · Authors · 2019-11-15
> **Comments to AnonReviewer 3**
>
> We appreciate the reviewer for the positive and valuable comments.
>
> >>> “It is not clear if the authors combined existing techniques and/or if they invented a new one.”
> A) Please note that our proposed approach in terms of both high-level idea and technical details is a non-trivial solution to solve a very challenging problem (e.g., inferring unknown subtask dependencies) that is relevant to real-world applications. Our MSGI model consists of adaptation policy, inductive logic programming (ILP) module, and test policy, and they operate in a single meta-learning framework. We used the existing CART with Gini impurity (Breiman et al., 2017) for our ILP module (see Figure 2) and the GRProp policy (Sohn et al., 2018) for test policy. Rather than in individual sub-modules, our main contribution lies in the whole MSGI framework, where we propose 1) to use a separate policy for adaptation and test phase to enable the agent to explore during adaptation and exploit during testing and 2) to use an ILP method to efficiently infer the task parameter (i.e., subtask graph) for faster adaptation. The list of contributions other than the model are summarized in the last paragraph of Section 1.
>
> >>> “What is the big difference from the work by Sohn et al. (2018)?”
> A) Sohn et al. (2018) require subtask graphs to be explicitly given to the agent to solve the task. However, our MSGI algorithm can solve the task without the requirement of subtask graph being given; instead, our method can “infer” the unknown subtask graph (per task) from the agent’s experience during adaptation. To make it even more clear, we clarified the difference in Section 2.2 as follows: “Our problem extends the subtask graph execution problem in (Sohn et al., 2018) by removing the assumption that a subtask graph is given to the agent; thus, the agent must infer the subtask graph in order to perform the complex task.”
>
> >>> “The authors evaluated one agent. It would have been better if authors trained multiple agents and showed a performance distribution (Fig 5.).”
> A) In fact, we reported the performance distribution over multiple runs in Figure 5. Specifically, we reported the mean (solid line) and standard error (shaded area) of the performance over 5 random seeds, 500 tasks (i.e., subtask graphs), and 32 test episodes. We note that the standard error (shaded area) is quite small in Figure 5, since we measured the performance over the large number of runs.
>
> Presentation:
>
> >>> “Figure 3 does not give a description of the subtask graph (middle) and the StarCraft II.”
> A) We added more description of the subtask graph and the task of SC2 in the caption of Figure 3 as follows: “...The goal is to execute subtasks in the optimal order to maximize the reward within time budget. The subtask graph describes subtasks with the corresponding rewards (e.g., transforming a chest gives 0.1 reward) and dependencies between subtasks through AND and OR nodes. For instance, the agent should first transform chest AND transform diamond before executing pick up duck.”
>
> >>> “Section 5.1.2 does not clearly explain the different datasets D1-D5 of Playground.”
> A) We added more detail about the Playground and Mining domain including explanation about D1-D4 and Eval datasets in Appendix C. We also added a pointer to Appendix C in Section 5.1.2. Thank you for pointing this out.

---

### Official Review · AnonReviewer1 · 2019-10-22
**Official Blind Review #1**

**Rating:** 6

**Review:**

This paper proposes a new meta-reinforcement learning algorithm, MSGI, which focuses on the problem of adapting to unseen hierarchical tasks through interaction with the environment where the external reward is sparse. The authors make use of subtask graph inference to infer the latent subtask representation of a task through interacting with the environment using an adaptation policy and then optimize the adaptation policy based on the inferred latent subtask structure. Each task in the paper is represented as a tuple of subtask precondition and subtask reward, which are inferred via logic induction and MLE of Gaussians respectively. At meta-test time, MSGI rollouts a subtask graph execution (SGE) policy based on the graph inferred from the interactions between the environment and the adaptation policy. The authors also propose a UCB-inspired intrinsic reward to encourage exploration when optimizing the adaptation policy. Experiments are conducted on two grid-world domains as well as StarCraft II.

Overall, this paper is mainly an extension of the prior work [1], which uses a subtask graph for tackling hierarchical RL problems. This work builds upon [1] by extending to meta-learning domains and studying generalization to new hierarchical tasks. While the contribution seems a bit incremental and the experimental setting is a bit unclear and limited to low-dimensional state space, the inference of task-specific subtask graphs based on past experiences and the proposal of a UCB-inspired reward shed some interesting insights on how to approach meta-hierarchical RL where long-horizon tasks and sparse rewards have been major challenges. Given some clarification on the experimental setup and additional results on more challenging domains in the author's response, I would be willing to improve my score.

Regarding the experimental setup, the set of subtasks is a Cartesian product of the set of primitive actions and a set of all types of interactive objects in the domain, while the state is represented as a binary 3-dimensional tensor indicating the position of each type of objects. Such a setup seems a bit contrived and is limited to low-dimensional state space and discrete action space, which makes me doubt its scalability to high-dimensional continuous control tasks. It would be interesting to see how/if MSGI can perform in widely used meta-RL benchmarks in Mujoco. I also wonder how MSGI can be compared to newly proposed context-based meta-RL methods such as PEARL.

As for the results, the authors don't provide an ablation study on the UCB exploration bonus though they claim they would show it in the paper. Moreover, the result of GRProp+Oracle is also missing in the comparison. I also don't understand why MSGI-Meta and RL2 would overfit in the SC2LE case and are unable to adapt to new tasks. Is that a limitation of the method? The authors also introduce MSGI-GRProp in this setting, which is never discussed before, and claim that MSGI-GRProp can successfully generalize to new tasks. It seems that the authors don't use a meta-RL agent in order to get this domain to work. I believe more discussion on this part is needed.

[1] Sungryull Sohn, Junhyuk Oh, and Honglak Lee. Hierarchical reinforcement learning for zero-shot
generalization with subtask dependencies. In NeurIPS, pp. 7156–7166, 2018.

**Experience Assessment:**

I have published one or two papers in this area.

**Review Assessment: Checking Correctness Of Derivations And Theory:**

N/A

**Review Assessment: Checking Correctness Of Experiments:**

I assessed the sensibility of the experiments.

**Review Assessment: Thoroughness In Paper Reading:**

I read the paper at least twice and used my best judgement in assessing the paper.

---

> ### Author Response · Authors · 2019-11-15
> **Comments to AnonReviewer 1**
>
> We appreciate the reviewer for their positive evaluation and detailed, constructive comments. We have updated the draft to address some concerns, and below we answer to the questions in more detail.
>
> >>> “Why MSGI-Meta and RL^2 would overfit in the SC2LE case and are unable to adapt to new tasks. Is that a limitation of the method?”
> A) The subtask graph structure (i.e., the Tech Tree) in SC2LE is fixed, as it is the inherent design of the game. For instance, Marine can only be built from Barrack and this remains fixed across the tasks. This limits the variation among the tasks, and the training and testing tasks in terms of the subtask graph structure become identical. In this case, the meta-learning method can achieve (near-) optimal performance by simply overfitting to the training tasks (i.e., memorizing the optimal sequence of options in training tasks) without any generalization over different subtask structures. Thus, this is not a limitation of our method, but a limitation of SC2LE domain in designing diverse tasks for meta-training.
>
> >>> “It seems that the authors don't use a meta-RL agent in order to get this domain (SC2LE) to work. I believe more discussion on this part is needed.”
> A)
> (1) Why we did not include any meta-RL agent in the SC2LE domain: As answered to the above comment, we did not include meta-RL agents since meta-training makes the problem too trivial in SC2LE domain as we cannot generate as many different subtask graphs as necessary for meta-learning to work.
> (2) How can our MSGI model work without a meta-policy:
> The ILP module infers the underlying subtask graph from the agent’s trajectory. In principle, however, our ILP module can infer the underlying subtask graph from any trajectory data (collected by any policy), as long as its coverage is enough. Our meta-trained policy makes this data collection as efficient as possible for more accurate inference.
> As shown in Figure 5, even the MSGI-Rand agent (i.e., our MSGI model but replacing the meta-policy with a random policy) outperform other baselines; it can still generate the experience data that can be used in ILP module.
>
> >>> “Additional results on more challenging domains”
> A) As an effort to make it more convincing that (1) our MSGI model can solve complex hierarchical task and that (2) real-world tasks have a hierarchical structure in it, we conducted an additional experiment on the AI2-THOR [1] environment. Here, we defined the cooking task similar to the breakfast preparation task described in the introduction with 148 realistic subtasks such as “slice bread” or “cook egg”. We added the details of the experiment and the result in the Appendix G. The experimental results on AI2-THOR show that our MSGI model can infer the underlying subtask graph accurately, and adapt more efficiently than the compared methods.
>
> >>> “The set of subtasks is a Cartesian product of the set of primitive actions and a set of all types of interactive objects in the domain.”
> A) We implemented a completion of the subtasks as a Cartesian product of the primitive actions and objects in Playground domain, but in general the completion set of a subtask can be any set of states as defined in Section 2.2. For example, the subtasks in SC2LE was defined by the design of the domain (i.e., high-level actions appearing in the environment’s API). Our model does not benefit from such a specific form, and in fact our MSGI can be applied as long as the completion and eligibility of subtask can be defined. To avoid confusion, we moved the Cartesian product-based definition to Appendix C, which explains the details of Playground and Mining domain, and provided the general definition of subtask in Appendix B.
>
> >>> “Such a setup (discrete subtasks and grid-based observation) seems a bit contrived and is limited to low-dimensional state space and discrete action space, which makes me doubt its scalability to high-dimensional continuous control tasks.”
> A)
> Since the policies in MSGI model operates on the option/subtask level, it can scale in terms of the number of subtasks. For both continuous and discrete state/action space cases, an MDP can be abstracted into discrete subtasks, and then our MSGI model can be applied to solve the task efficiently. In such an environment with high-dimensional observation spaces, the policy can make use of both raw, high-dimensional observations (to learn a mapping/relation between subtasks and parts of the observation) and the discrete subtask information. We also note that our experiment shows that MSGI scales well to large number of subtasks (e.g., 148 subtasks for AI2-THOR [1]) and high-dimensional observation space (e.g., SC2LE and AI2-THOR).

---

> ### Author Response · Authors · 2019-11-15
> **Comments to AnonReviewer 1 (Cont'd)**
>
>
> >>> “It would be interesting to see how/if MSGI can perform in widely used meta-RL benchmarks in Mujoco” --- MSGI on the standard tasks without hierarchy?
> A) In the standard benchmark tasks without hierarchy (i.e., single-goal), the performance of MSGI solely depends on the performance of the option since the task consists of a single subtask. Because our work assumes that the option is given, the problem becomes trivial; thus, we did not include the standard tasks without hierarchy in this paper. We instead focus on the complex hierarchical tasks that existing HRL and meta-RL methods cannot solve efficiently.
>
> >>> “As for the results, the authors don't provide an ablation study on the UCB exploration bonus though they claim they would show it in the paper.”
> A) We added the ablation study result in Appendix H. Figure 17 shows that UCB exploration bonus term helps meta-training in Playground and Mining domain. Thanks for pointing this out.
>
> >>> “Moreover, the result of GRProp+Oracle is also missing in the comparison”
> A) GRProp+Oracle was used as the upper bound for performance normalization of all the agents (Section 5.1). Therefore, in Figure 4 and 5, $\widehat{R}=1$ corresponds to the performance of GRProp+Oracle. To avoid confusion, we made it more clear in the comment of Figure 5 by adding “The performance of each method was normalized where GRProp+Oracle is the upper bound (i.e., $\widehat{R}=1$) and Random is the lower bound (i.e., $\widehat{R}=0$).”
>
> >>> “The authors also introduce MSGI-GRProp in this setting, which is never discussed before, and claim that MSGI-GRProp can successfully generalize to new tasks.”
> A) Our MSGI agent consists of two policies: adaptation policy and test policy. MSGI-GRProp uses the GRProp [2] policy as an adaptation policy (instead of meta-learned adaptation policy) while other parts are unchanged. We made it more clear in Section 5.2.
> References:
> [1] Kolve et al., AI2-THOR: An Interactive 3D Environment for Visual AI, ArXiv, 2017

---

### Official Review · AnonReviewer2 · 2019-10-22
**Official Blind Review #2**

**Rating:** 6

**Review:**

Summary
-------------
The authors propose a novel meta-rl problem where hierarchical tasks are characterized by a graph describing all sub-tasks and their dependencies. They propose a meta-rl approach to meta-train a policy that quickly infers the subtask graph from new task data. The approach is compared to relevant baselines from both the meta-rl and hierarchical rl literature on complex domains. In particular, the authors consider a large-scale Startcraft II experiment which proves the efficiency and scalability of the proposed methodology.

Major Comments
--------------

Meta-rl is a relevant direction for reducing the sample-complexity of rl agents and scaling them to large domains. This work presents interesting and novel ideas in these settings. In particular, the few-shot rl problem with subtask dependencies seems quite interesting for both encoding and solving large hierarchical rl problems. The proposed meta-rl algorithm is sound and simple to understand. The paper is well-organized, though sometimes it is difficult to follow the formalisms due to a large number of different symbols introduced. The experiments are quite interesting and convincing. In particular, the Starcraft domain should address all concerns about the scalability and efficiency of the proposed approach. Some comments/questions follow.

1. The state available to the agent includes the number of remaining time-steps and episodes. When/how are they used?

2. The paper requires the reader to be quite familiar with some previous works (e.g., Section 3.2 requires to know Song et al. 2018 to understand the test phase). It would be good to add more background/details about these works (at least in the supplementary), so that the paper is more self-contained.

3. In the Starcraft experiment, what is the difference between MSGI-meta and MSGI-GRProp? Furthermore, where is the "oracle" baseline (introduced in sec. 5) used in the experiments? I did not find any plot reporting it.

4. The main limitation is that this approach requires options for each subtask to be provided before-hand. Do the authors think that the method is easily generalizable to learn such options as well? Furthermore, I realized this limitation only after reading the very last lines of the paper. Since this is of major importance, I believe it should be clearly stated much earlier.

Minor Comments
--------------
1. First line of sec. 2.1: R_\tau should be R_G
2. I did not find a definition of o_t and d_t which appear, e.g., in Algorithm 1.

**Experience Assessment:**

I have published one or two papers in this area.

**Review Assessment: Checking Correctness Of Derivations And Theory:**

I assessed the sensibility of the derivations and theory.

**Review Assessment: Checking Correctness Of Experiments:**

I assessed the sensibility of the experiments.

**Review Assessment: Thoroughness In Paper Reading:**

I read the paper at least twice and used my best judgement in assessing the paper.

---

> ### Author Response · Authors · 2019-11-15
> **Comments to AnonReviewer 2**
>
> We appreciate the reviewer for the constructive and helpful comments.
>
> >>> 1. “When/how are the number of remaining time-steps and episodes used?”
> A) Intuitively, the agent will only execute the subtasks that can be executed within the remaining time step to get the reward, since the agent receives the reward corresponding to each subtask only if it “finishes” the subtask before episode terminates. Also, these two time features (i.e., remaining time-steps and episodes) need to be given to the agent for time-awareness [1] when the MDP is of finite-horizon. In our implementation, these time features were given to the policy as additional inputs.
>
> >>> 2. “It would be good to add more background/details about the previous works (e.g., [Sohn et al., 2018]) that this paper relies on (at least in the supplementary).”
> A) In Appendix A, we added more details of backgrounds in [Sohn et al., 2018] that are relevant to this paper. We added an explicit reference to this in Section 2.
>
> >>> 3-1. “In the SC2 experiment, what is the difference between MSGI-meta and MSGI-GRProp?”
> A) MSGI-meta uses a meta-learned policy as an adaptation policy, while MSGI-GRProp uses GRProp policy. Motivation of MSGI-GRProp: GRProp is a good approximation/heuristic algorithm that works well without meta-training as shown in [Sohn et al., 2018]. We have made this motivation more clear in Section 5.2 as follows: “... Instead of MSGI-Meta, we used MSGI-GRProp. MSGI-GRProp uses the GRProp policy as an adaptation policy, since GRProp is a good approximation algorithm that works well without meta-training as shown in (Sohn et al., 2018).”
>
> >>> 3-2. “Where is the "oracle" baseline (introduced in sec. 5) used in the experiments?”
> A) GRProp+Oracle was used as the upper bound for performance normalization of all the agents. So, in Figure 4 and 5, $\widehat{R}=1$ corresponds to the performance of GRProp+Oracle. We made it more clear in the comment of Figure 5 by adding “The performance of each method was normalized where GRProp+Oracle is the upper bound (i.e., $\widehat{R}=1$) and Random is the lower bound (i.e., $\widehat{R}=0$).”
>
> >>> 4-1. “The limitation of assuming options to be given can be stated much earlier.”
> A) In the footnote 1 (in Section 2.2) of the submission, we stated that we assume options are pre-learned. We will move this to the main text to make this more clear.
>
> >>> 4-2. Can we learn options and subtasks instead of assuming that it is given?
> We believe this is possible but a highly non-trivial problem. One possible way is to use the option discovery methods [2, 3]. Consider the option that executes a subtask. The completion and eligibility set corresponds to the “termination condition” and “initiation set” in the option framework, which can be learned [4]. It is however not directly applicable to our framework since the ILP module requires the perfect (i.e., noise-free) completion and eligibility input, which is hard to achieve with the learned eligibility and completion predictor. Thus we leave it as a future work.
>
>
> Minor typos:
>
> A) Thank you for finding the typos. We corrected the typo in Section 2.1 and added the missing definitions in Section 2.2.
>
> References:
> [1] Pardo, Fabio, et al. Time limits in reinforcement learning. arXiv, 2017. https://arxiv.org/abs/1712.00378
> [2] Krishnan et al., DDCO: Discovery of Deep Continuous Options for Robot Learning from Demonstrations, CoRL 2017. https://arxiv.org/abs/1710.05421
> [3] Ramesh et al., Successor Options: An Option Discovery Framework for Reinforcement Learning https://arxiv.org/abs/1905.05731
> [4] Harutyunyan et al., The Termination Critic, https://arxiv.org/abs/1902.09996

---

> > ### Comment · AnonReviewer2 · 2019-11-15
> > **Post-rebuttal comments**
> >
> > Thank you for the detailed response. The authors addressed all of my questions and updated the paper accordingly. I thus confirm my initial view and vote for acceptance.

---

### Decision · Program_Chairs · 2019-12-19

**Decision:**

Accept (Poster)

**Comment:**

This work formulates and tackles a few-shot RL problem called subtask graph inference, where hierarchical tasks are characterized by a graph describing all subtasks and their dependencies. In other words, each task consists of multiple subtasks and completing a subtask provides a reward. The authors propose a meta-RL approach to meta-train a policy that infers the subtask graph from any new task data in a few shots. Empirical experiments are performed on different domains, including Startcraft II, highlighting the efficiency and scalability of the proposed approach.

Most concerns of reviewers were addressed in the rebuttal. The main remaining concerns about this work are that it is mainly an extension of Sohn et al. (2018), making the contribution somewhat incremental, and that its applicability is limited to problems where subtasks are provided. However, all reviewers being positive about this paper, I would still recommend acceptance.